# A non-genetic switch triggers alternative telomere lengthening and cellular immortalization in ATRX deficient cells

Timothy K. Turkalo[1,8], Antonio Maffia[1,8], Johannes J. Schabort[1], Samuel G. Regalado[1], Mital Bhakta[2], Marco Blanchette[2], Diana C. J. Spierings [3], Peter M. Lansdorp [4,5] & Dirk Hockemeyer [1,6,7] ✉

Alternative Lengthening of Telomeres (ALT) is an aberrant DNA recombination pathway which grants replicative immortality to approximately 10% of all cancers. Despite this high prevalence of ALT in cancer, the mechanism and genetics by which cells activate this pathway remain incompletely understood. A major challenge in dissecting the events that initiate ALT is the extremely low frequency of ALT induction in human cell systems. Guided by the genetic lesions that have been associated with ALT from cancer sequencing studies, we genetically engineered primary human pluripotent stem cells to deterministically induce ALT upon differentiation. Using this genetically defined system, we demonstrate that disruption of the p53 and Rb pathways in combination with ATRX loss-of-function is sufficient to induce all hallmarks of ALT and results in functional immortalization in a cell type-specific manner. We further demonstrate that ALT can be induced in the presence of telomerase, is neither dependent on telomere shortening nor crisis, but is rather driven by continuous telomere instability triggered by the induction of differentiation in ATRX-deficient stem cells.

Overcoming proliferative mortality is a hallmark of cancer development[1]. For most normal somatic tissues, proliferative capacity is restricted by the progressive loss of telomere repeats at the end of every chromosome with each round of cell division[2,3]. In about 10% of cancers, this proliferation barrier is overcome by activating a mechanism termed Alternative Lengthening of Telomeres (ALT), a recombination-based process which diverts the cell's inherent homology-directed repair (HDR) machinery to extend telomeres and expand its proliferative capacity[4,5]. Immortalization through the ALT pathway is associated with decreased long-term patient survival compared to cancers that immortalize by expressing the telomere synthesizing enzyme telomerase[6–8].

Approximately 90% of ALT cancers harbor mutations in ATRX or DAXX[9–11], which deposit histone H3.3 in heterochromatic genomic regions, including telomeres[12–16]. In addition, ALT cancer cells share several distinct molecular hallmarks including heterogeneous and often very long (>50 kb) telomeres. Telomeric DNA frequently colocalizes with promyelocytic leukemia (PML) protein in the nucleus, forming ALT-associated PML bodies (APBs)[17]. ALT, but not telomerase-positive cancer cells, generate extrachromosomal partially single-stranded telomeric $(CCCTAA)_n$ DNA circles referred to as C-circles[18]. ALT shares key features with the break-induced replication (BIR) recombination mechanism[19–22], leading to telomere synthesis outside of S-phase and increased rates of telomere sister chromatid exchange

[1]Department of Molecular and Cell Biology, University of California, Berkeley, CA 94720, USA. [2]Dovetail Genomics, Enterprise Way, Scotts Valley, CA, USA. [3]European Research Institute for the Biology of Ageing, University of Groningen, University Medical Centre Groningen, Groningen, The Netherlands. [4]Terry Fox Laboratory, BC Cancer Agency, Vancouver, BC V5Z 1L3, Canada. [5]Department of Medical Genetics, University of British Columbia, Vancouver, BC V6T 1Z4, Canada. [6]Chan Zuckerberg Biohub, San Francisco, CA 94158, USA. [7]Innovative Genomics Institute, University of California, Berkeley, CA 94720, USA. [8]These authors contributed equally: Timothy K. Turkalo, Antonio Maffia. ✉e-mail: hockemeyer@berkeley.edu

(T-SCE)[23]. Finally, inactivation of the DNA-sensing cGAS-STING pathway has been shown to be associated with ALT cancers, likely as a means to evade autophagy triggered by the activation of the innate immune system pathway[24,25].

Several models of how ATRX suppresses ALT have been proposed, including promoting sister telomere cohesion[26,27], suppressing aberrant DNA structures[28], promoting faithful replication and repair[29,30], maintaining proper chromatin accessibility[31], and preventing RNA-DNA hybrid-based instability[32]. Loss of ATRX and DAXX can also be found in telomerase positive cancers, albeit with low frequency[33,34], and some ALT phenotypes have been observed in telomerase-positive cancer lines that have acquired ATRX mutations[35]. While ATRX has been identified as the most common mutation in ALT cancers, its knockout in SV40 large T antigen (SV40 LT)-transformed primary fibroblasts only increased the frequency of immortalization via ALT by about two-fold over those with wildtype ATRX or DAXX[36–38], implying that additional changes are required to induce ALT and immortalize cells.

Here we sought to test this hypothesis by generating a genetically tractable system engineered to carry ALT-predisposing mutations identified by cancer GWAS before testing possible epigenetic triggers for ALT. For this purpose, ATRX was genetically inactivated in human pluripotent stem cells which are deficient in the cell cycle checkpoint and DNA damage response genes $p16$ and $TP53$. In the pluripotent state, ATRX deficient cells mostly lack ALT characteristics and maintain telomere length and genome integrity like typical telomerase-positive stem cells. Upon differentiation and exit from the pluripotent state, these ATRX knockout cells rapidly acquired the key features of ALT, including ALT-like long and heterogeneous telomeres. Such cells proliferated past the point where contemporaneous and isogenic ATRX proficient cells entered crisis. These results demonstrate that non-genetic changes that occur as a consequence of the differentiation of ATRX deficient cells can drive telomere recombination and cellular immortalization.

## Results

### ATRX loss significantly increases ALT immortalization frequency

To identify the required genetic or epigenetic changes that generate immortalized cells via ALT, we engineered an $ATRX$ knockout (ATRX$^{-/-}$) in human embryonic stem cells (hESCs) by deleting exon 1 of $ATRX$ in a $CDKN2A$ exon 2 deficient genetic background that abrogates p16 and p14 function (Supplementary Fig. 1a and Supplementary Table 1 lists the cell lines in this study). Deletion of $ATRX$ exon 1 resulted in elimination of ATRX protein expression (Supplementary Fig. 1b). Attempts to mutate $ATRX$ in checkpoint proficient cells failed to produce any knockout clones (Supplementary Tables 2 and 3). The ATRX$^{-/-}$ hESCs remained pluripotent based on the expression of the pluripotency marker OCT4 (Supplementary Fig. 1c), though they had a shorter mean telomere length than the isogenic wildtype control cells and grew slightly slower in culture (Fig. 1c and Supplementary Fig. 1d). In this isogenic cell system, we asked whether loss of ATRX would result in immortalized ALT clones. Differentiation of both $ATRX$ wildtype (ATRX$^{+/+}$) and ATRX$^{-/-}$ hESCs into telomerase negative fibroblasts resulted in a loss of OCT4 expression as expected (Supplementary Fig. 1c). However, specifically ATRX deficient but not wildtype cells abruptly stopped proliferating shortly after differentiation (Supplementary Fig. 1e). To recapitulate studies which assessed immortalization rates in SV40 LT-transformed fibroblasts, we transduced these ATRX$^{-/-}$ fibroblasts with a lentivirus encoding SV40-IRES-RFP immediately after differentiation induction, when cell proliferation was stagnant (Fig. 1a and Supplementary Fig. 1e). Despite a low titer of transduction and lack of selection, many clones (~100–200) grew as RFP-positive colonies. We established clonal lines and kept the remaining colonies as a bulk culture. Surprisingly, the SV40 LT-transformed ATRX$^{-/-}$ fibroblast

clones and the bulk culture showed all key features of ALT cells at or exceeding the level of the established ALT osteosarcoma line U2OS (Fig. 1b–g). The SV40 LT-treated ATRX$^{-/-}$ fibroblasts, but not the ATRX$^{+/+}$ cells, exhibited APBs (Fig. 1b, d and Supplementary Fig. 1g, h), heterogeneous long telomeres (Fig. 1c), EdU incorporation at telomeres outside of S-phase (Fig. 1e) and C-circles (Fig. 1f). Moreover, a significant fraction of telomeres in these cells were recognized as sites of DNA damage indicated by the presence of telomere dysfunction-induced foci (TIFs) as previously reported for ALT cells[38] (Fig. 1g and Supplementary Fig. 1i). Based on the RFP infection efficiency, a conservative estimate is that >1% of the transduced cells continue to proliferate and acquire the hallmarks of ALT (>100 colonies arising from $<1 \times 10^4$ cells infected with SV40-IRES-RFP). This efficiency exceeds previous reports by more than 1000-fold. We continuously passaged the resulting cells for >120 days (>60 PD) in culture, indicating that they had indeed immortalized.

### Differentiation of ATRX$^{-/-}$ hESCs is sufficient to induce the molecular hallmarks of ALT

To separate the impacts of stem cell differentiation and loss of checkpoint on the appearance of ALT phenotypes we next assessed the acute effects of differentiation on the induction of ALT features without SV40 LT transduction. To rapidly trigger differentiation, we passaged hESCs as single cells at low density in feeder-free hESC medium lacking the pluripotency factor TGFβ: we term this medium E7[39]. We compared ATRX$^{+/+}$ and ATRX$^{-/-}$ hESCs grown on MEFs to their differentiated counterparts. ATRX$^{-/-}$ hESCs did not exhibit any ALT phenotypes (Fig. 2a–f). In contrast, E7-differentiated ATRX$^{-/-}$ cells upregulated expression of APBs and EdU incorporation at telomeres (Fig. 2a-c), exhibited a modest increase in C-circle expression (Fig. 2d and Supplementary Fig. 1j) and had a significant increase in TIFs upon differentiation (Fig. 2e). After only one week of differentiation, telomeres became long and heterogeneous exclusively in differentiated ATRX$^{-/-}$ cells, indicative of ALT activation (Fig. 2f). Consistent with the DNA damage signal at telomeres and despite acquiring the hallmarks of ALT, differentiated ATRX$^{-/-}$ cells ceased proliferation as observed earlier (Fig. 1a and Supplementary Fig. 1e). In agreement with our SV40 LT data, transduction of these ATRX$^{-/-}$ fibroblasts with Cas9 targeting $TP53$ rescued this proliferation defect (Fig. 2g). We conclude that differentiation of ATRX deficient cells leads to an acute induction of DNA damage and heterogeneous length of telomeres, but continued proliferation is only possible by loss of p53 activity. This observation agrees with ALT cancer data: $TP53$ mutations are common in ALT cancers, Li-Fraumeni patient fibroblasts (congenital $TP53$ mutation) more readily immortalize via ALT[4], and $TP53$ mutations are slightly predictive for the ALT telomere maintenance mechanism over telomerase[40].

### ALT activation is independent of telomere crisis but requires differentiation

So far our data suggests that differentiation can rapidly trigger ALT phenotypes, an observation that is in contrast to the previous view that ALT is the consequence of the selection taking place once cells' telomeres become short and cells enter crisis[37,38]. If this were true, passaging ATRX deficient, checkpoint deficient, TERT-negative hESCs into crisis should result in ALT-immortalized cells. To directly test this hypothesis, we built an hESC system that allows us to control TERT expression and engineered ALT-associated mutations into this cell line. We used a previously characterized cell line where TERT is overexpressed from the AAVS1 locus flanked by loxP sites in order to be able to tightly control the onset of crisis (Fig. 3a)[41,42]. We further introduced homozygous endogenous knockouts of p53 and p16 to render cells checkpoint-deficient without the need for SV40 transduction (Supplementary Figs. 2a–l and 3a). We will refer to the resulting $ATRX^{-/-}$, $TP53^{-/-}$, $p16^{-/-}$, $TERT^{-/-}$

*AAVS1: TERT*[c/c] as ATRX[−/−] TERT[c/c] hESCs and the isogenic ATRX unedited clones as ATRX[+/+], TERT[c/c] cells (Supplementary Table 1 lists the genotypes used in this study). *TP53*[+/+] and *CDKN2A*[+/+] hESCs failed to produce any ATRX[−/−] TERT[c/c] clones (Supplementary Table 3). Deletion of *ATRX* exon 1 (Supplementary Fig. 2j–l), confirmed our previous observation with cells having significantly shorter telomeres and slower proliferation compared to wildtype hESCs

(Supplementary Fig. 3b–d). Cells did not show induction of a global DNA damage response by checkpoint activation (Supplementary Fig. 3e) and telomeres did not show telomere dysfunction induced foci (TIF) (Supplementary Fig. 3f), suggesting that they are not detected as sites of DNA damage. ATRX[−/−] TERT[c/c] hESCs had a significantly increased C-circle signal compared to ATRX[+/+] TERT[c/c] but no significant increase in telomeric EdU incorporation or APB

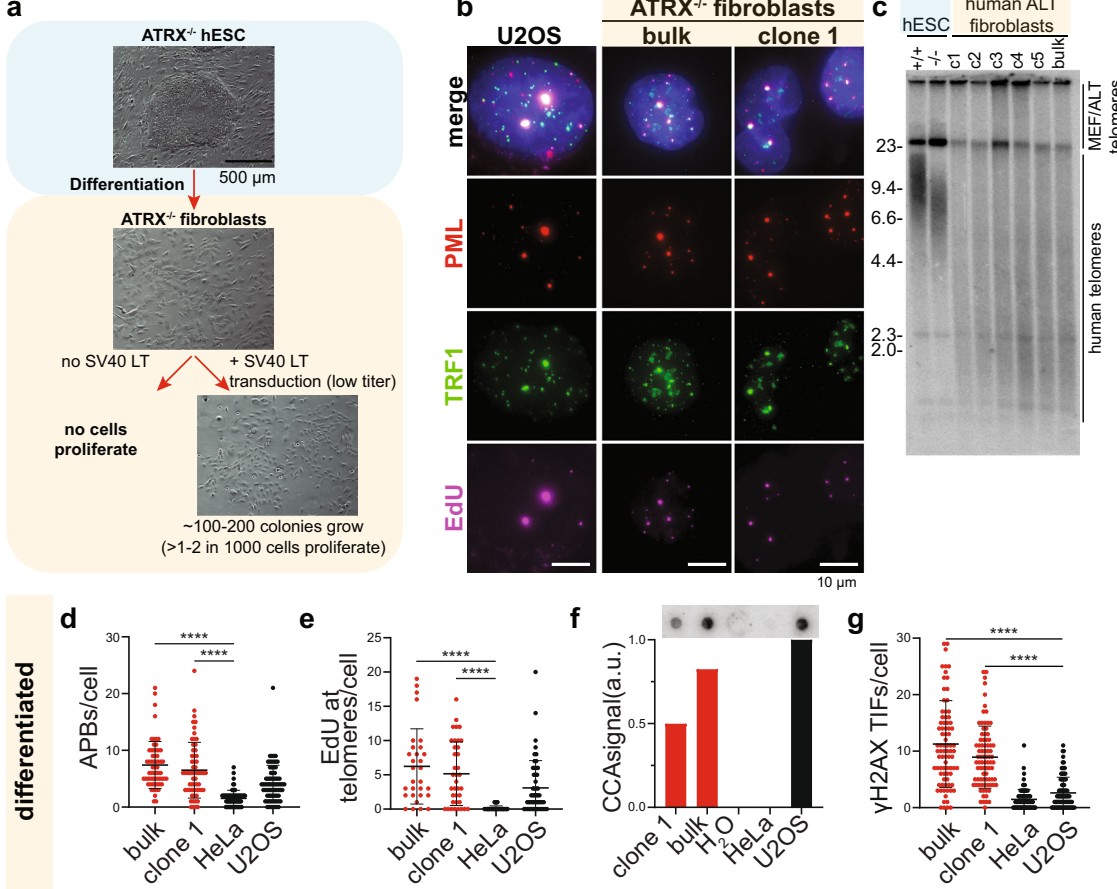

**Fig. 1 | ALT induction in ATRX[−/−] cells. a** Experimental overview of ATRX[−/−] hESC differentiation and SV40-LT immortalization. ATRX was genetically ablated in hESCs, then cells were differentiated into fibroblasts (see "Methods"). Early after differentiation (20 days) cells were either mock transduced or infected with RFP-SV40 LT. Transduced cells proliferated and the efficiency of survival was estimated by counting surviving fibroblast colonies. **b** Representative immunofluorescence images of APB staining (see "Methods"). Cells were simultaneously stained for PML (red), TRF1 (green), and EdU (violet), with DAPI (blue) counterstain for nuclei. Cells were assayed shortly after differentiation (7 days) and treated with 10 μM RO-3306 24 h before fixation. 2 h prior to fixation, 10 μM EdU was added to the media. U2OS cells were used as positive control and show large APBs due to PML, TRF1 and EdU colocalization. Both bulk culture of ATRX[−/−] fibroblasts and a representative isolated clone show colocalization of PML, TRF1 and EdU (quantified in **d**). Images are here represented as maximum intensity projection. Scale bar is 10 μm. **c** Telomere restriction fragment (TRF) blot comparing hESCs (ATRX[+/+] and ATRX[−/−]) and ATRX[−/−] differentiated fibroblasts. hESCs show a more discrete telomere length. It is worth noticing that ATRX[−/−] cells show a shorter telomere length setpoint. ATRX[−/−] isolated clones and bulk culture show ALT typical heterogeneous lengths appearing as a smear detected by radioactive telomeric probes. DNA ladder sizes are reported along the gel in kilobases. **d** Immunofluorescence analysis of PML/TRF1 colocalizations (APBs). Colocalizations were counted per each nucleus (DAPI). HeLa and U2OS (black dots) are used as negative and positive controls, respectively. Cells were assayed shortly after differentiation (7 days). ATRX[−/−] bulk culture and a representative clonal line (red) show a significant increase in APBs/cell when compared to the controls. ATRX[−/−] bulk: *n* = 61; ATRX[−/−] clone 1: *n* = 60; HeLa: *n* = 66; U2OS: *n* = 73 over 1 experiment. Data shown are individual values with means ± s.d.,

asterisks represent *p* value (*p* < 0.0001) as calculated by Kruskal−Wallis test. Source data are provided as a Source data file. **e** Immunofluorescence analysis of EdU/TRF1 colocalizations. Colocalizations were counted per each nucleus (DAPI). Cells were assayed shortly after differentiation (7 days). HeLa and U2OS (black dots) are used as negative and positive controls respectively. ATRX[−/−] bulk culture and a representative clonal line (red) show a significant increase in EdU/telomere per cell when compared to the controls. ATRX[−/−] bulk: *n* = 30; ATRX[−/−] clone 1: *n* = 37; HeLa: *n* = 55; U2OS: *n* = 52 over 1 experiment. Data shown are individual values with means ± s.d., asterisks represent *p* value (*p* < 0.0001) as calculated by Kruskal−Wallis test. Source data are provided as a Source data file. **f** Quantification of C-circle assay. The assay was performed in HeLa and U2OS cells as negative and positive controls respectively (black bar). In addition, $H_2O$ is used as reaction negative control. ATRX[−/−] bulk fibroblast culture and a representative clonal line (red bars) show an increase in C-circle signal when compared to controls. C-circles signal intensity was detected after hybridization (see "Methods"), corrected on Alu signal intensity and normalized for U2OS signal (=1), *n* = 1. Source data are provided as a Source data file.
**g** Immunofluorescence analysis of γH2AX/TRF1 colocalizations (TIFs). Colocalizations were counted per each nucleus (DAPI). Cells were assayed shortly after differentiation (7 days). HeLa and U2OS (black dots) are used as negative and positive controls respectively. ATRX[−/−] fibroblast bulk culture and a representative clonal line show a significant increase of TIFs per cell when compared to controls. ATRX[−/−] bulk: *n* = 83; ATRX[−/−] clone 1: *n* = 83; HeLa: *n* = 116; U2OS: *n* = 87 over 1 experiment. Data shown are individual values with means ± s.d., asterisks represent *p* value (*p* < 0.0001) as calculated by Kruskal−Wallis test, *n* = 1. Source data are provided as a Source data file.

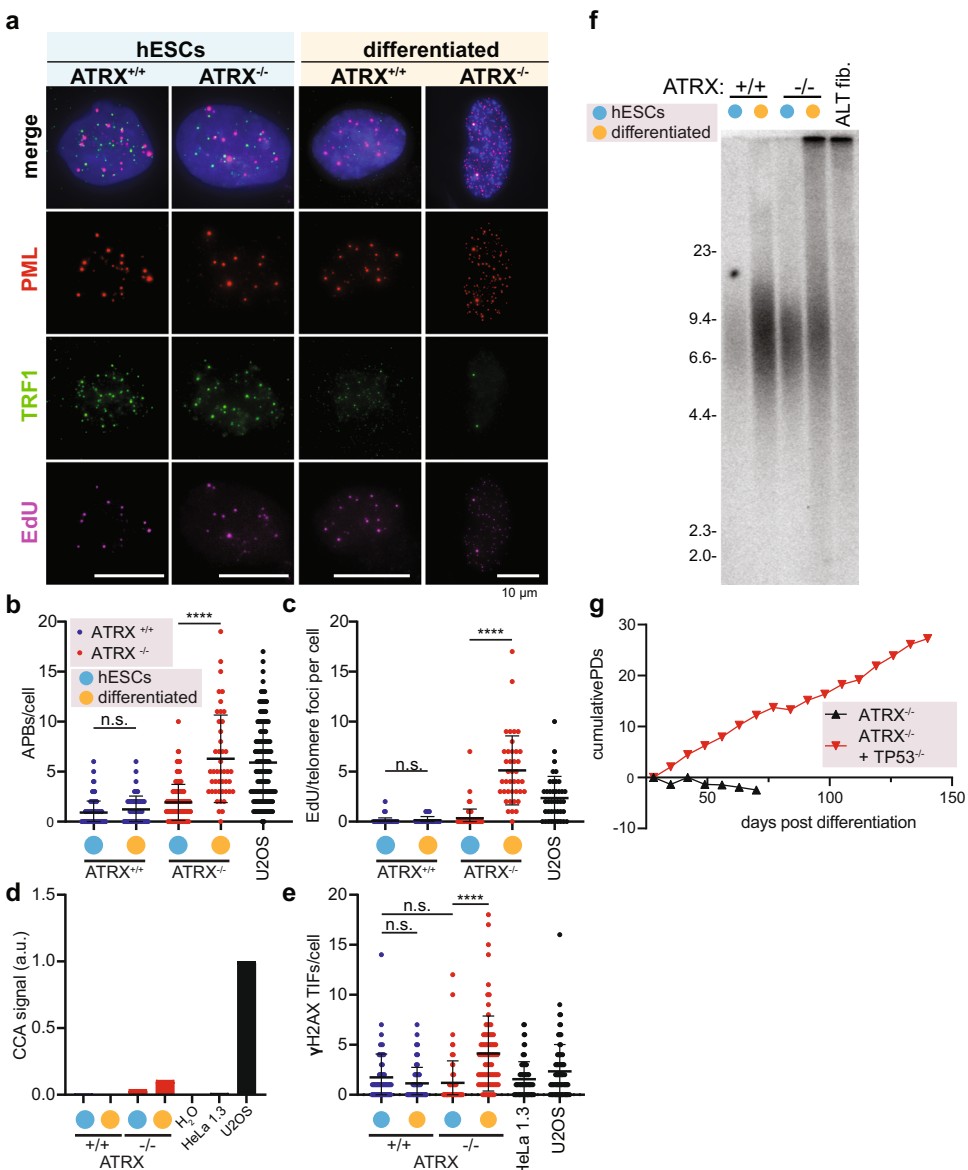

formation (Fig. 3b–g and Supplementary Fig. 3g, h). The observation that C-circles were absent or low in ATRX⁻/⁻, CDKN2A⁻/⁻ hESCs, yet robustly detectable in independent ATRX⁻/⁻, TERTᶜ/ᶜ clones suggests a specific role of p53 in the accumulation of C-circles.

ATRX⁻/⁻ hESCs showed slower proliferation after Cre-mediated loopout of TERT from the AAVS1 locus (Supplementary Fig. 3i). Telomeres in the ATRX⁻/⁻, TERT⁻/⁻ hESCs shortened as expected when compared to the parental ATRX⁻/⁻, TERTᶜ/ᶜ cells (Fig. 3h). Serial passage of both ATRX⁻/⁻, TERT⁻/⁻ hESCs as well as the ATRX⁺/⁺, TERT⁻/⁻ control cells resulted in crisis and no colonies emerged. Analysis of 5 independent ATRX⁻/⁻, TERT⁻/⁻ and 3 ATRX⁺/⁺, TERT⁻/⁻ cell lines revealed that the proliferative capacity of cells after TERT loopout correlated with the starting telomere length (Fig. 3i and Supplementary Fig. 3b, c). Single telomere length analysis (STELA) confirmed that ATRX⁺/⁺, TERT⁻/⁻ and ATRX⁻/⁻, TERT⁻/⁻ hESCs stopped proliferating at a similar telomere length despite their different proliferative capacities (Supplementary Fig. 3j). Collectively, these experiments show that continuously passaging the same number of hESCs (approximately 5–10 × 10⁶) into telomere crisis does not lead to the recovery of ALT positive immortalized hESC clones. TRF and STELA analysis of DNA samples collected shortly before the cultures stopped proliferating suggest that telomeres in both ATRX⁺/⁺ and ATRX⁻/⁻ cells are critically short as previously documented in TERT⁻/⁻ hESC lines (Supplementary Fig. 3j).

The lack of APBs and EdU incorporation at telomeres in the ATRX⁻/⁻ stem cell state was in striking contrast to when we differentiated the very same ATRX⁻/⁻ TERTᶜ/ᶜ cells into fibroblasts. Both ATRX⁻/⁻, TERTᶜ/ᶜ and ATRX⁻/⁻, TERT⁻/⁻ fibroblast lines upregulated the ALT phenotype, while the isogenic ATRX⁺/⁺, TERTᶜ/ᶜ and ATRX⁺/⁺, TERT⁻/⁻ cells did not (Fig. 3b–f and Supplementary Fig. 4a–i). APBs and EdU incorporation at telomeres were detected in ATRX⁻/⁻, TERTᶜ/ᶜ and ATRX⁻/⁻, TERT⁻/⁻ as early as cells could be sampled. In addition, telomeres became long and heterogeneous similarly to ALT telomeres only in the fibroblast state (Fig. 3h and Supplementary Fig. 4j). Because these phenotypes were observed in both the telomerase negative and telomerase overexpressing cells, we conclude that cells do not need to enter telomere crisis to initiate ALT recombination. In fact, TERT over-expression is fully compatible with the key features of ALT and loss of telomerase activity is not necessary for the activation of ALT in our isogenic cell system.

To generalize that differentiation induces ALT from ATRX⁻/⁻ TERTᶜ/ᶜ hESCs, we also differentiated cells into other cell types.

**Fig. 2 | Differentiation of ATRX⁻/⁻ hESCs is sufficient to induce the molecular hallmarks of ALT. a** Representative immunofluorescence images of APB staining (see "Methods"). Matched hESCs and E7 differentiated cells were stained simultaneously for PML (red), TRF1 (green), and EdU (violet), with DAPI (blue) counterstain for nuclei. Cells were assayed shortly after differentiation (7 days), treated with 10 μM RO-3306 for 24 h and 10 μM EdU was added to the media 2 h prior to fixation, U2OS cells were used as positive control and show large APBs due to PML, TRF1 and EdU colocalization. Only ATRX⁻/⁻ differentiated cells show colocalization of PML, TRF1 and EdU as opposed to hESCs or the differentiated ATRX⁺/⁺ control. Scale bar is 10 μm. **b** Immunofluorescence quantification of PML/TRF1 colocalizations (APBs) per nuclei (DAPI) in ATRX⁺/⁺ (blue) and ATRX⁻/⁻ (red), hESCs (cyan dots) and matched E7 cultures (yellow dots). ATRX⁺/⁺ cells (blue) do not show presence of APBs/cells even after differentiation by E7 protocol (see "Methods"). ATRX⁻/⁻ cells differentiated by E7 protocol show a significant increase of APBs/cell when compared to the corresponding hESCs grown on MEFs (see "Methods") ($p < 0.0001$). U2OS cells (black) are also represented as positive control. ATRX⁺/⁺: $n = 163$ for hESCs, $n = 58$ for differentiated cells; ATRX⁻/⁻: $n = 158$ for hESCs, $n = 45$ for differentiated cells; U2OS: $n = 111$ over 1 experiment. Data shown are individual values with means ± s.d., asterisks represent $p$ value ($p < 0.0001$) comparing ATRX⁻/⁻ cells on MEFs culture vs differentiation as calculated by Kruskal–Wallis test. Source data are provided as a Source data file. **c** Immunofluorescence quantification of EdU/TRF1 colocalizations per nuclei (DAPI) in ATRX⁺/⁺ (blue) and ATRX⁻/⁻ (red) hESCs (cyan dots) and matched E7 cultures (yellow dots). ATRX⁺/⁺ cells (blue) do not show presence of EdU at telomeres even after differentiation by E7 protocol (see "Methods"). ATRX⁻/⁻ cells differentiated by E7 protocol show a significant increase of EdU/telomere when compared to the corresponding hESCs grown on MEFs ($p < 0.0001$). U2OS cells (black) are also represented as positive control. ATRX⁺/⁺: $n = 132$ for hESCs, $n = 48$ for differentiated cells; ATRX⁻/⁻: $n = 95$ for hESCs, $n = 40$ for differentiated cells; U2OS: $n = 40$ over 1 experiment. Data shown are individual values with means ± s.d., asterisks represent $p$ value ($p < 0.0001$) comparing

ATRX⁻/⁻ cells on MEFs culture vs differentiation as calculated by Kruskal-Wallis test. Source data are provided as a Source data file. **d** Quantification of C-circle assay in hESCs (cyan dots) and following 1 week of E7 differentiation (yellow dots). ATRX⁻/⁻ cells (red bars) show an increase in C-circle signal after differentiation when compared to ATRX⁺/⁺ (here undetectable) and HeLa 1.3 as negative control. U2OS cells (black bar) have been used as positive control. H₂O has been used as reaction negative control. Source data are provided as a Source data file. **e** Quantification of the immunofluorescence staining of γH2AX/TRF1 colocalizations (TIFs) per nuclei (DAPI). ATRX⁺/⁺ cells (blue) show no increase of TIFs after differentiation. ATRX⁻/⁻ cells (red) show a significant increase of TIFs per cell after E7 differentiation when compared to the hESCs state ($p \leq 0.0001$). HeLa 1.3 and U2OS (black dots) are used as negative and positive controls respectively. ATRX⁺/⁺: $n = 63$ for hESCs, $n = 81$ for differentiated cells; ATRX⁻/⁻: $n = 74$ for hESCs, $n = 81$ for differentiated cells; HeLa: $n = 92$; U2OS: $n = 95$ over 1 experiment. Data shown are individual values with means ± s.d., asterisks represent p-value ($p < 0.0001$) comparing ATRX⁻/⁻ cells on MEFs culture or differentiation as calculated by Kruskal–Wallis test. Source data are provided as a Source data file. **f** Telomere restriction fragment (TRF) assay of ATRX⁺/⁺ and ATRX⁻/⁻ hESCs and matched E7 cultures 1 week after differentiation, compared to ALT positive fibroblasts previously isolated. ATRX⁺/⁺ cells show no difference in telomere length after differentiation, while E7 differentiated ATRX⁻/⁻ cells show a smeared telomeric signal comparable to the previously generated ALT-positive fibroblasts. DNA fragment sizes are indicated along the gel in kilobases. **g** Growth curve reporting cumulative population doublings (PDs) plotted for as long as 150 days. ATRX⁻/⁻ differentiated cells were transduced with a lentiviral construct expressing Cas9 along with a sgRNA targeting TP53 (red line) (see "Methods"). Knockout of p53 allowed ATRX⁻/⁻ differentiated cells to continue proliferating for as long as 150 days. p53-positive uninfected cells (black line) were depleted from the culture shortly after differentiation. Source data are provided as a Source data file.

---

Differentiation into neuronal precursor cells (NPCs) and E7 resulted in APBs in ATRX⁻/⁻ cells (Supplementary Fig. 5a, b). Differentiation in E7 also resulted in upregulation of APBs at telomeres in all ($n = 10$) ATRX⁻/⁻ clones compared to ATRX⁺/⁺ control cells (Supplementary Fig. 5c). Importantly, for ATRX⁻/⁻ TERTᶜ/ᶜ cells, but not ATRX⁺/⁺ TERTᶜ/ᶜ cells, differentiation into NPCs or fibroblast like cells resulted in the establishment of long and heterogeneous telomeres compared to contemporaneous hESC cultures (Supplementary Fig. 5d). We conclude that differentiation, and therefore a non-genetic change in ATRX⁻/⁻ cells, can induce the rapid appearance of the ALT phenotypes.

## Differentiation of ATRX⁻/⁻ hESCs results in immortal ALT cells
Next, we asked if the induction of the key ALT features in ATRX deficient cells results in telomere maintenance and functional immortalization. To test this, we removed TERT by Cre-mediated loopout in the stem cell state, then differentiated the cultures (Fig. 4a). ATRX⁻/⁻ hESCs differentiated by both fibroblast and E7 protocols showed a significant increase in APB formation and EdU incorporation at telomeres (Fig. 4b–d). Again, ATRX⁻/⁻, TERT⁻/⁻ cultures proliferated slower than ATRX⁺/⁺ cultures (Fig. 4e), similar to the stem cell state (Supplementary Fig. 3i) and underwent a phase of decreased proliferation. Of note, differentiated ATRX⁻/⁻, TERTᶜ/ᶜ had long and heterogenous telomeres, had telomerase activity and could be continuously maintained in culture (Supplementary Fig. 4j, k). ATRX⁺/⁺, TERT⁻/⁻ cells failed to immortalize when continuously passaged as undifferentiated hESCs and when differentiated. In contrast, ATRX⁻/⁻, TERT⁻/⁻ differentiated cultures recovered and continued to proliferate. Interestingly, differentiated ATRX⁻/⁻, TERT⁻/⁻ cells slowed their proliferation at around 150 days but then eventually kept dividing for more than 250 days (Fig. 4e). This recovery in proliferation rate cannot be attributed merely to telomere length changes, as telomeres in ATRX⁻/⁻, TERT⁻/⁻ cells became heterogeneous in length shortly after differentiation and remained largely stable over time with a significant fraction of telomeres being very short (Fig. 4f, g). This suggests that additional

factors, other than telomere length, contribute to the efficient immortalization of these cells.

## Ongoing telomere recombination drives early genomic instability in ALT cells
So far, our data has shown that loss of ATRX in a permissive cellular state renders telomeres as recombinogenic sites of DNA damage, independent of telomere length. This finding suggests a model in which the genomic instability seen in ALT is not caused by telomere shortening preceding ALT pathway activation but rather that ATRX loss in differentiated cells induces ongoing genomic instability that limits cellular proliferation in these incipient ALT cells.

To test this hypothesis, we compared the genome integrity role of ATRX in stem cells and in differentiated cells after ALT induction. We used Strand-seq[43], a technique that detects structural chromosome aberrations and measures the frequency of sister chromatid exchanges (SCEs) (Fig. 5 and Table 1)—an intrachromosomal equivalent to sister telomere exchanges found in ALT cells (see below). To exclude effects of telomere shortening and crisis, we performed this analysis in TERTᶜ/ᶜ cells. We included RTEL1⁻/⁻ and BLM⁻/⁻ knockout cells as positive controls (Supplementary Fig. 6a–c)[44,45]. Wildtype and ATRX⁻/⁻, TERTᶜ/ᶜ hESCs did not have significantly different numbers of SCEs, which were detected in the BLM⁻/⁻ positive control (Fig. 5 and Supplementary Fig. 6d–h). Ploidy was also largely maintained in ATRX⁻/⁻, TERTᶜ/ᶜ hESCs compared to ATRX⁺/⁺, TERTᶜ/ᶜ (Fig. 5), while the RTEL1⁻/⁻ hESC line was highly aneuploid (Supplementary Fig. 6e). In contrast to the relatively stable genome in the stem cell state, an increase in copy number alterations was observed in the differentiated ATRX⁻/⁻, TERTᶜ/ᶜ cells resulting in significant heterogeneity of ploidies between individual cells (Fig. 5 and Table 1) indicating ongoing genomic instability. Interestingly, despite this genomic instability detectable in differentiated ATRX⁻/⁻, TERTᶜ/ᶜ cells, we only detected a very minor increase in SCEs (Supplementary Fig. 6f). We conclude that ATRX knockout in hESCs does not cause major intra-genomic instability, but that

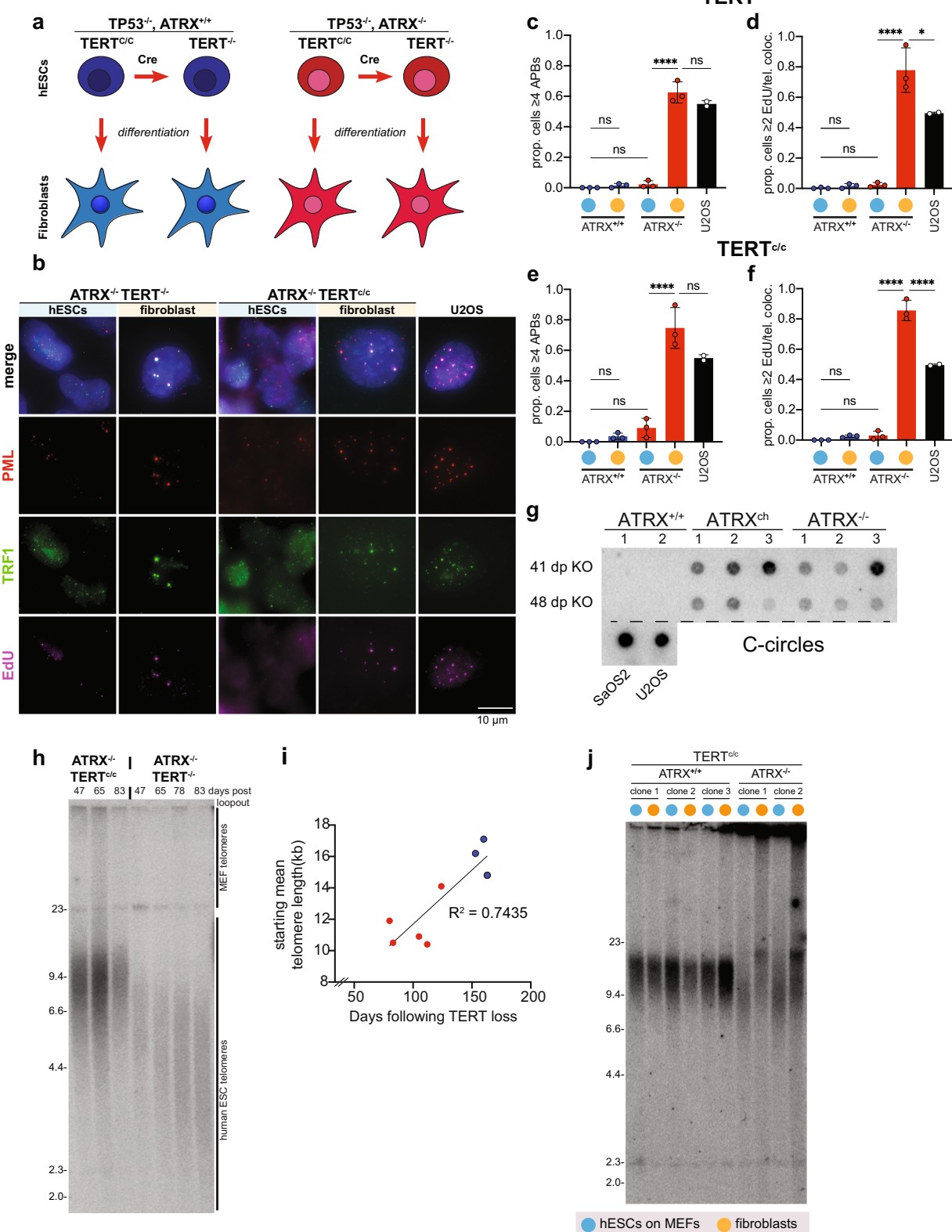

activation of ALT following differentiation promotes genome rearrangements.

To establish the basis of the genomic instability caused after the induction of ALT, we decided to closely monitor the chromosomal events that occur after immortalization. We characterized the genomes of immortalized subclones derived from ATRX⁻/⁻, TERT⁻/⁻ cells approximately 260 days in culture after TERT loopout and used

ATRX⁺/⁺, TERT⁻/⁻ cells 75 days post TERT loopout as a comparison (see Fig. 4e). The earlier timepoint for ATRX⁺/⁺, TERT⁻/⁻ cells was chosen to enable subcloning and expansion of sufficient cells for our analysis before the cells entered crisis and ceased proliferating. In addition to APBs and incorporation of EdU at telomeres, all ATRX⁻/⁻, TERT⁻/⁻ clones had telomeres of heterogeneous length and still retained a substantial fraction of short telomeres despite immortalization

**Fig. 3 | Crisis does not activate ALT in hESCs. a** Overview of the TERT Cre-mediated loopout in hESCs prior to fibroblasts differentiation (see "Methods"). ATRX$^{-/-}$ and ATRX$^{+/+}$ TERT$^{c/c}$ hESCs were transfected with Cre mRNA or mock treated and then differentiated into fibroblasts. **b** Representative immuno-fluorescence images of APB staining (see "Methods"). Matched hESCs and fibroblast differentiated cells were stained simultaneously for PML (red), TRF1 (green), and EdU (violet), with DAPI (blue) counterstain for nuclei. Cells were assayed shortly after differentiation, treated with 10 μM RO-3306 for 24 h and 10 μM EdU was added to the media 2 h prior to fixation. U2OS cells were used as positive control and show numerous APBs due to PML, TRF1 and EdU colocalization. ATRX$^{-/-}$ differentiated cells show APBs irrespectively of TERT status. Colocalization foci are not present in the hESC state. Scale bar is 10 μm. **c** Proportion of APBs-positive TERT$^{-/-}$ cells calculated as ≥4 APBs/cell in hESCs (cyan dots) and fibroblasts (yellow dots). ATRX$^{+/+}$ cells (blue bars) show no significant increase of APBs in any cell state. ATRX$^{-/-}$ cells (red bar) show a significant increase of APBs per cell when differentiated ($p < 0.0001$). The level of APBs is comparable to U2OS cells used as positive control (black bar). Prop≥4: the proportion of cells in the total population with ≥4 APBs per cell. Graph represents the mean ± s.d. of three independent clones plotted as dots, asterisks represent $p$ value as calculated by ordinary one-way ANOVA. Source data are provided as a Source data file. **d** Proportion of EdU/TRF1 colocalization-positive TERT$^{-/-}$ cells calculated as ≥2 EdU foci/telomere in hESCs (cyan dots) and fibroblasts (yellow dots). ATRX$^{+/+}$ cells (blue bars) show no significant increase of EdU/telomere in any cell state. ATRX$^{-/-}$ cells (red bar) show a significant increase of EdU/telomere per cell when differentiated ($p < 0.0001$). The level of EdU is significantly higher than U2OS cells used as positive control (black bar). Prop≥2: the proportion of cells in the total population with ≥2 EdU foci per cell. Graph represents the mean ± s.d. of three independent clones plotted as dots, asterisks represent p-value as calculated by ordinary one-way ANOVA. Source data are provided as a Source data file. **e** Proportion of APBs-positive TERT$^{c/c}$ cells calculated as ≥4 APBs/cell in hESCs (cyan dots) and fibroblasts (yellow dots). ATRX$^{+/+}$ cells (blue bars) show no significant increase of APBs in any cell state.

ATRX$^{-/-}$ cells (red bar) show a significant increase of APBs per cell when differentiated ($p < 0.0001$). The level of APBs is comparable to U2OS cells used as positive control (black bar). Prop≥4: the proportion of cells in the total population with ≥4 APBs per cell. Graph represents the mean ± s.d. of three independent clones plotted as dots, asterisks represent $p$ value as calculated by ordinary one-way ANOVA. Source data are provided as a Source data file. **f** Proportion of EdU/TRF1 colocalization-positive TERT$^{C/C}$ cells calculated as ≥2 EdU foci/telomere in hESCs (cyan dots) and fibroblasts (yellow dots). ATRX$^{+/+}$ cells (blue bars) show no significant increase of EdU/telomere in any cell state. ATRX$^{-/-}$ cells (red bar) show a significant increase of EdU/telomere per cell when differentiated ($p < 0.0001$). The level of EdU is significantly higher than U2OS cells used as positive control (black bar). Prop≥2: the proportion of cells in the total population with ≥2 EdU foci per cell. Graph represents the mean ± s.d. of three independent clones plotted as dots, asterisks represent $p$ value as calculated by ordinary one-way ANOVA. Source data are provided as a Source data file. **g** C-circle assay of different ATRX$^{+/+}$, ATRX$^{ch}$ (compound heterozygous) and ATRX$^{-/-}$ clones. SaOS and U2OS cells are used as positive controls in the assay. Signal was quantified and normalized on Alu probe and SaOS signal (see Supplementary Fig. 3g). **h** Telomere restriction fragment assay of ATRX$^{-/-}$ hESCs before Cre-mediated loopout of TERT (TERT$^{c/c}$) and after loopout (TERT$^{-/-}$) days after TERT loopout are reported. ATRX$^{-/-}$ cells show the appearance of heterogeneous telomere fragments after TERT loss and a shortening of telomeres overtime. DNA fragment sizes are indicated along the gel in kilobases. **i** Quantification of mean telomere length (kb) plotted versus days to crisis following Cre transfection. ATRX$^{+/+}$ and ATRX$^{-/-}$ clones are represented as blue and red dots respectively. The graph highlights how the initial telomere length of each clone correlates with the time to crisis after TERT loss. Source data are provided as a Source data file. **j** Telomere restriction fragment assay of different clonal lines of TERT$^{c/c}$ hESCs (cyan dots) and differentiated fibroblasts (yellow dots). ATRX$^{+/+}$ clones retain a discrete telomere length in any cell state. ATRX$^{-/-}$ clones show the appearance of heterogeneous telomere fragments after differentiation in fibroblasts. DNA fragment sizes are indicated along the gel in kilobases.

(Fig. 6a and Supplementary Fig. 7a–c). This recapitulates the telomere length distributions previously observed in the bulk population (Fig. 4g). In contrast to the bulk analysis however, ATRX$^{-/-}$, TERT$^{-/-}$ subclones and U2OS cells revealed clonal-specific stabilized TTAGGG repeats as detected by telomeric probes indicative of internal telomeric repeats resulting from telomere fusion events[46]. Analysis of metaphase spreads by telomere FISH and telomere CO-FISH confirmed the presence of telomere-telomere fusions and a high level of T-SCEs (Fig. 6b, c, Supplementary Fig. 7d, e, and Supplementary Table 4). Moreover, we analyzed these clones for TBK1 phosphorylation, which serves as a downstream sensor of cGAS-STING pathway activation[47,48]. In agreement with previous reports, we find that pTBK1 is reduced in U2OS and the ATRX$^{-/-}$, TERT$^{-/-}$ subclones compared to HeLa and ATRX$^{+/+}$, TERT$^{-/-}$ subclones (Supplementary Fig. 7f). This observation further confirms that attenuation of the cGAS-STING pathway is a requirement for ALT immortalization[24,25]. Together, these findings led us to hypothesize that aberrant telomeric recombination events precede and might drive the early stages of cellular immortalization via ALT. To map these aberrant chromosomal structural events unbiasedly, comprehensively and at higher resolution, we performed a Micro-C assay in all the ATRX$^{+/+}$, TERT$^{-/-}$ and ATRX$^{-/-}$, TERT$^{-/-}$ clones together with U2OS cells. We detected numerous chromosomal terminal contacts in ATRX$^{-/-}$, TERT$^{-/-}$ differentiated clones and U2OS cells but not in the ATRX$^{+/+}$, TERT$^{-/-}$ cells (Fig. 6d, Supplementary Fig. 8, and Supplementary Table 5). Analysis of common terminal contacts between different ATRX$^{-/-}$, TERT$^{-/-}$ clones allowed us to recreate a hierarchy of reoccurring genomic events and provided a key insight into the effects of ALT immortalization on genome integrity (Fig. 6e). We identified a series of both terminal and internal contacts that are common to all clones, representing the ancestral events that occurred early after differentiation.

In addition, consecutive events involving the chromosome termini were characteristic of individual ATRX$^{-/-}$, TERT$^{-/-}$ clones, indicating that induction of ALT is concurrent with ongoing genomic

instability driven by telomere recombination. Continuous aberrant telomere rearrangements generate dicentric chromosomes and the signature of breakage-fusion-bridge cycles[49–51] yet occur long after cells have induced telomere maintenance. Thus, our data argues that incipient ALT cancer cells need to select for genomic alterations that balance the need for telomere-telomere recombination with its adverse effects on genome instability.

## Discussion

### Non-genetic changes in cellular states can be a key driver of ALT

The recent discovery of TERT promoter mutations has dramatically increased our understanding about the steps that lead to cellular immortality of cancer cells by telomerase reactivation[52–56]. A similar genetic roadmap for how cells activate ALT, the second mechanism that cancers deploy to overcome replicative senescence, has been lacking. Here, we describe a genetically controlled system that recapitulates the genetic steps associated with ALT and thereby resolves the order of events and requirements for ALT dependent immortalization. Specifically, we demonstrate that the knockout of ATRX is only permissive in checkpoint deficient hESCs, yet not sufficient to induce the molecular signatures of ALT cells: EdU incorporation at telomeres outside S-phase, APBs formation, and telomere maintenance. Consequently, checkpoint deficient ATRX and TERT double knockout hESCs undergo progressive telomere shortening and fail to immortalize. This highlights how telomeres shortening and crisis are insufficient to directly trigger ALT.

Our key insight into the ALT mechanism came after differentiating the same ATRX$^{-/-}$ hESCs to somatic cell types. Shortly after differentiation these cells exhibited all the well-recognized ALT features and successfully immortalized. The fast kinetics with which differentiation induced ALT features in most cells cannot be explained by selection for additional mutations or by bulk telomere length changes but rather must be attributed to a faster switch controlling telomere recombination. Future experiments are needed to identify the molecular

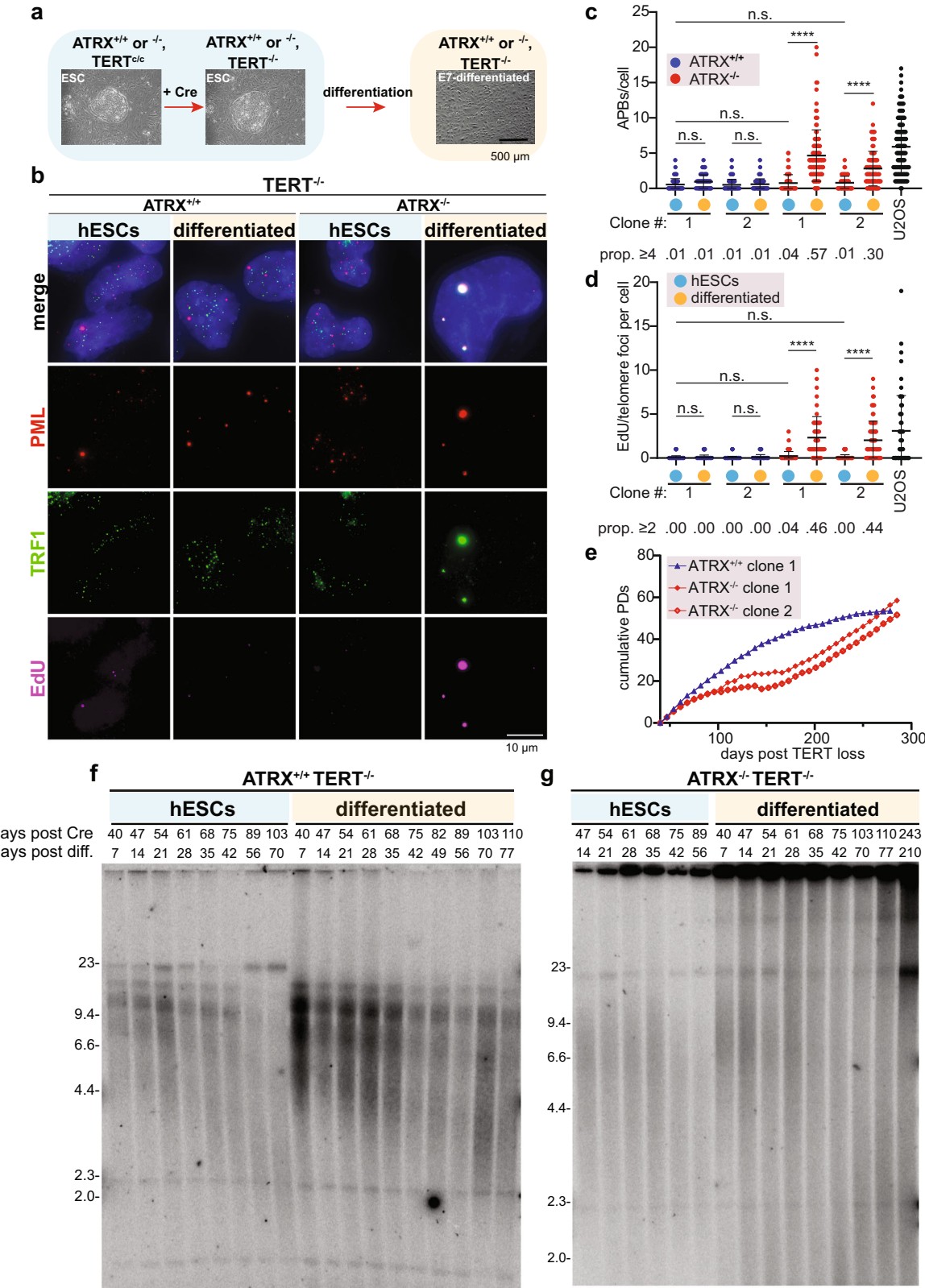

events that facilitate these changes. Considering the lack of ALT phenotypes in ATRX deficient dermal fibroblasts prior to crisis and their low frequency of spontaneous immortalization[37], we propose that, when present, ATRX marks telomeres as anti-recombinogenic by deposition of repressive chromatin marks. Thus, telomere status becomes self-maintained and irreversible. Our data further argues that telomeres are in a privileged stage in stem cells as they are resistant to

telomere-telomere recombination even after loss of ATRX. This is similar to the stem cell specific telomere status recently proposed for mouse ESCs, showing that in mouse pluripotent stem cells lacking a main component of the shelterin complex TRF2, telomere fusion is repressed while T-loops can still be formed[57,58]. The differentiation of TRF2 deficient cells results in loss of end deprotection, activation of the NHEJ pathway and telomere-telomere fusions. This parallel

**Fig. 4 | Differentiation of ATRX$^{-/-}$ hESCs results in functionally immortal ALT cells. a** Schematic overview of TERT Cre-mediated loopout in hESCs. TERT$^{C/C}$ hESCs, (either ATRX$^{+/+}$ or ATRX$^{-/-}$) were transfected with Cre mRNA, then differentiated by growth in E7 medium (see "Methods"). Examples of hESCs colonies (light blue panel) or differentiated cells (light yellow panel) are shown. **b** Representative immunofluorescence image of APBs staining in TERT$^{-/-}$ cells. ATRX$^{+/+}$ or ATRX$^{-/-}$ hESCs or E7 differentiated cells were stained for PML (red), TRF1 (green), and EdU (violet), with DAPI (blue) counterstain. Colocalization of APBs and EdU is evident in TERT$^{-/-}$, ATRX$^{-/-}$ cells only after differentiation (rightmost column). Maximum intensity projections are here represented. Scale bar is 10 μm. **c** Quantification of PML/TRF1 colocalizations (APBs) per cell for TERT$^{-/-}$ cells. ATRX$^{+/+}$ (blue) and ATRX$^{-/-}$ (red), hESCs (cyan dots) and matched E7 cultures (yellow dots). ATRX$^{+/+}$ cells show no significant increase of APBs/cell in either differentiation state. Differentiated ATRX$^{-/-}$ cells show a significant increase of APBs/cell when compared to the hESC matched culture. U2OS cells (black) are used as positive control. Prop≥4: the proportion of cells in the total population with ≥4 APBs per cell. ATRX$^{+/+}$ clone 1: $n = 124$ for hESCs, $n = 126$ for differentiated cells; ATRX$^{+/+}$ clone 2: $n = 109$ for hESCs, $n = 130$ for differentiated cells; ATRX$^{-/-}$ clone 1: $n = 118$ for hESCs, $n = 125$ for differentiated cells; ATRX$^{-/-}$ clone 2: $n = 118$ for hESCs, $n = 103$ for differentiated cells; U2OS: $n = 111$ over 1 experiment. Graph represents the mean ± s.d. of two independent clones, asterisks represent $p$ value ($p < 0.0001$) as calculated by Kruskal–Wallis test. Source data are provided as a Source data file. **d** Quantification of EdU/TRF1 colocalizations per cell for TERT$^{-/-}$ cells. ATRX$^{+/+}$ (blue) and ATRX$^{-/-}$ (red), hESCs (cyan dots) and matched E7 cultures (yellow dots). ATRX$^{+/+}$ cells show no significant increase of EdU/telomere in either differentiation state. Differentiated ATRX$^{-/-}$ cells show a significant increase of EdU/telomere when compared to the hESC matched culture. U2OS cells (black) are used as positive control. Prop≥2: the proportion of cells in the total population with ≥2 EdU foci per cell. ATRX$^{+/+}$ clone 1: $n = 103$ for hESCs, $n = 105$ for differentiated cells; ATRX$^{+/+}$ clone 2: $n = 112$ for hESCs, $n = 51$ for differentiated cells; ATRX$^{-/-}$ clone 1: $n = 110$ for hESCs, $n = 52$ for differentiated cells; ATRX$^{-/-}$ clone 2: $n = 98$ for hESCs, $n = 97$ for differentiated cells; U2OS: $n = 97$ over 1 experiment. Graph represents the mean ± s.d. of two independent clones, asterisks represent $p$ value ($p < 0.0001$) as calculated by Kruskal–Wallis test. Source data are provided as a Source data file. **e** Growth curve of E7 cultures, cumulative population doublings (PDs) are here plotted (see "Methods") along days after TERT Cre-loopout. ATRX$^{+/+}$ cells (blue line) reach a plateau phase while two different ATRX$^{-/-}$ clones overcome a lag phase and successfully immortalize as shown by linear increase of population doublings well over 250 days past TERT loss (red). Source data are provided as a Source data file. **f** Telomere restriction fragment blot of matched ATRX$^{+/+}$ hESC and cells differentiated by E7 protocol. Days after Cre-loopout and days after differentiation are reported. ATRX$^{+/+}$ cells telomeres shorten after E7 differentiation. DNA fragment sizes are indicated along the gel in kilobases. **g** Telomere restriction fragment blot of matched ATRX$^{-/-}$ hESC and cells differentiated by E7 protocol. Days after Cre-loopout and days after differentiation are reported. ATRX$^{-/-}$ cells show heterogeneous telomere lengths after differentiation. DNA fragment sizes are indicated along the gel in kilobases.

phenomenon highlights how embryonic stem cells have developed a privileged mechanism for telomere protection that dramatically changes during differentiation.

## The ALT cancer spectrum

ALT occurs in a specific cancer spectrum in terms of age and tissue specificity. ALT immortalized cancers are more frequently represented in pediatric and mesenchymal cancers. Despite this specificity, the mutations found in ALT cancers cannot be used to unequivocally define the genetic drivers as is seen for other cancers such as the Wnt signaling or the MAPK pathway driving intestinal cancer or melanoma respectively. A cell type specific non-genetic switch can explain the lower correlation between genetic mutations and ALT phenotype. We demonstrate that loss of ATRX and the p53/p16 DNA damage and cell cycle checkpoints are permissive to induce ALT in a cell state specific manner and so we can reconcile this observation. We propose that ALT arises from a precursor cell in which ATRX is required to prevent telomere recombination. An interesting parallel in this regard are mesenchymal cancers that are driven by onco-histones[59]. Mutations in histone H3.3 have an overlapping tumor spectrum with ALT cancer including a strong association with undifferentiated sarcomas and pediatric gliomas[60]. Probing these molecular parallels will require the generation of a detailed map of the epigenetic events that telomeres undergo during cellular differentiation as well as testing the causality of these changes regarding the suppression of telomere recombination.

## ATRX loss drives genomic instability upon differentiation

A key insight gained from our experiments is that induction of ALT features can be experimentally uncoupled from cellular immortalization. We demonstrate that in embryonic stem cells in which telomere length is stably maintained, ALT features are suppressed, and cells maintain genome stability. Shortly after differentiation, telomere lengths become highly heterogeneous in a telomerase independent manner, possibly resulting from different resolution of recombination intermediates. Despite activation of all key features of ALT, including telomere maintenance, differentiated ATRX$^{-/-}$, TERT$^{-/-}$ cells transition through a lag phase during immortalization that cannot be fully explained by telomere length since resulting immortalized clones still retain some very short telomeres. While this lag phase may also select for additional mutations which enable immortalization such as the

attenuation of the cGAS-STING pathway[24,25], our data demonstrate a clear impact of genomic instability during this time of early immortalization. By tracing telomere fusion events, we demonstrate that cells are subjected to genomic instability driven by telomere dysfunction and fusion events. We therefore propose that at these early steps of cancer development, an incipient cancer cell activating ALT is challenged to balance telomere recombination to gain immortality with concomitant genome instability. Recent reports support that ALT cells have specific adaptations to attenuate responses to extrachromosomal DNA including C-circles by inactivating the STING pathway[24,25] or counteract DNA replication stress and facilitate the resolution of complex recombination intermediates by co-opting specific DNA repair complexes[20,61].

In summary we propose that in the stem cell state, cells can rely on a yet to be clearly defined telomere protection mechanism that does not require ATRX to prevent telomeres from recombining. However, once cells differentiate and silence telomerase expression, ATRX is essential to mark telomeres so that they undergo telomere shortening and eventually senesce without using recombination to regain telomere repeats (Fig. 7). In the absence of ATRX, stem cells can differentiate into a recombinogenic-permissive state and trigger the alternative lengthening mechanism.

## Methods

### Tissue culture methods

Genome editing was performed in WIBR3 hESCs, NIH stem cell registry #0079, RRID:CVCL_9767[62]. Cell culture was carried out as previously described[63]. Briefly, hESC lines were maintained on a monolayer of CD-1 strain mouse embryonic fibroblasts (MEFs) (Charles River) inactivated by 35 Gy of γ-irradiation. hESCs were grown in hESC medium (DMEM/F12 (Gibco) supplemented with 20% KnockOut Serum Replacement (Gibco), 1 mM glutamine (Sigma-Aldrich), 1% non-essential amino acids (Gibco), 0.1 mM β-mercaptoethanol (Sigma-Aldrich), 100 U/mL Penicillin–Streptomycin (Gibco), and 4 ng/mL FGF-Basic (AA 1–155) (Gibco)). Cultures were passaged every 5–7 days either manually or enzymatically with 1.5 mg/mL collagenase type IV (Gibco) by sedimentation and washing 3 times in wash medium (DMEM (Gibco) supplemented with 5% newborn calf serum (Sigma-Aldrich) and 100 U/mL Penicillin–Streptomycin (Gibco)).

HeLa 1.3 cervical carcinoma cells were a gift of T. de Lange (The Rockefeller University, New York, NY)[64]. U2OS and Saos-2

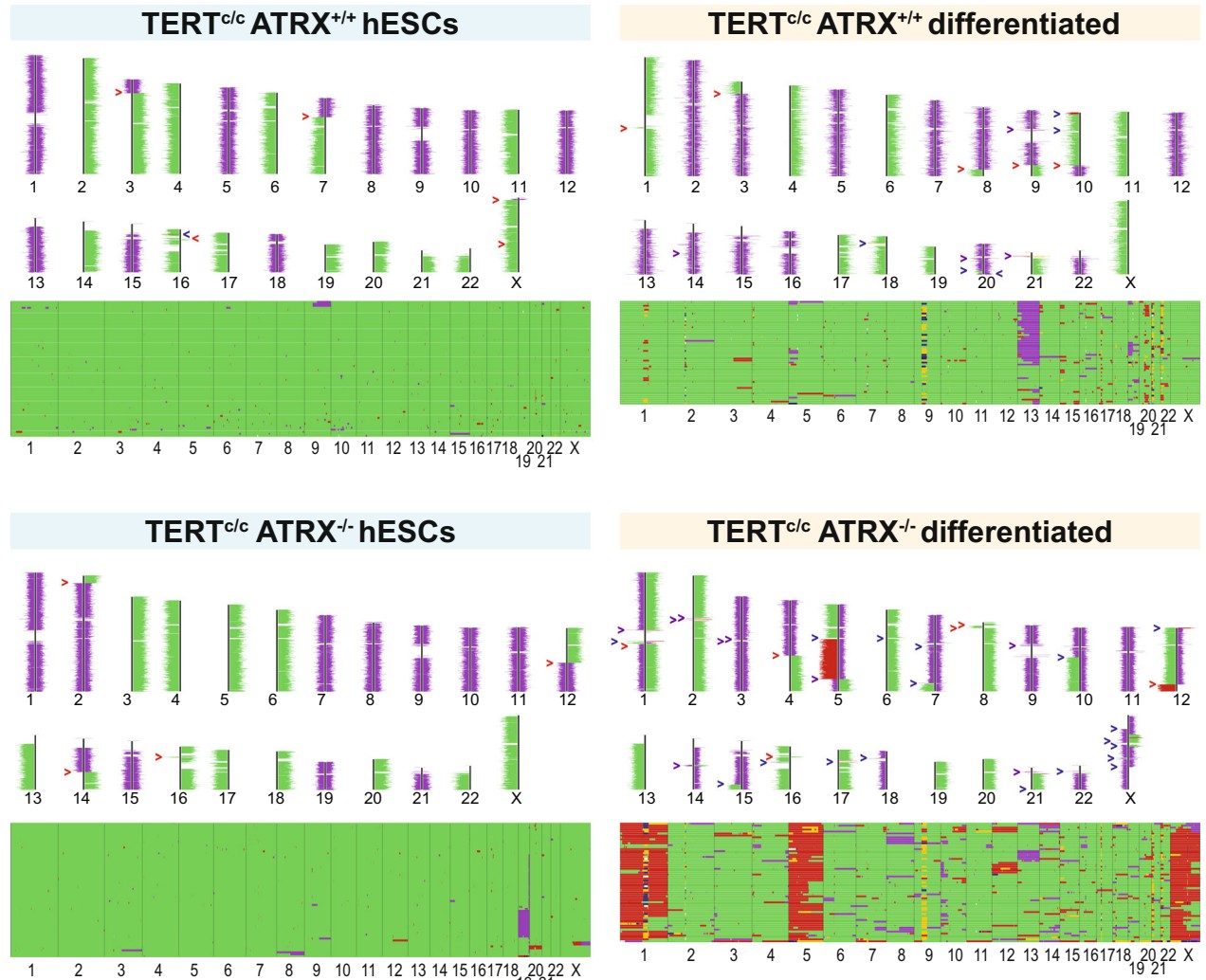

**Fig. 5 | Differentiation of ATRX[−/−] hESCs results in significant ongoing genome instability.** Examples of results from single cell Strand-seq analysis. Reads from individual Strand-seq libraries were mapped to the positive ("Crick") or negative ("Watson") strand of the human reference genome. The number of reads per binned genome interval was used to calculate ploidy (purple: haploid; green: diploid; red trisomy using Aneufinder[73]). Upper panels: reads mapping to Watson/Crick strands in selected individual cells. Red arrowheads identify copy number neutral template switch, blue arrowheads point at copy number alterations and purple is used for copy number alterations resulting in template switch. Lower panels, horizontal lines: ploidy per cell. Analysis of sister chromatid exchanges in upper panels shows no significant differences in the hESC state. Trisomies of chr 1, 5 and X are observed in ATRX[−/−] differentiated cells. In addition, many cells carry additional non-clonal numerical and segmental chromosome aberrations, which indicates ongoing genomic/chromosomal instability.

**Table 1 | Genome instability measured by Strand-seq**

|                        | Aneuploidy | Heterogeneity |
|------------------------|------------|---------------|
| ATRX[+/+] hESC          | 0.01       | 0.01          |
| ATRX[−/−] hESC          | 0.01       | 0.02          |
| ATRX[+/+] differentiated | 0.09       | 0.09          |
| ATRX[−/−] differentiated | 0.29       | 0.22          |
| RTEL1[−/−] hESC         | 0.54       | 0.26          |
| BLM[+/+] hESC           | 0.04       | 0.07          |
| BLM[−/−] hESC           | 0.12       | 0.10          |

Quantification of results from single-cell Strand-seq analysis obtained using Aneufinder[73], see "Methods" section for a more detailed description of the analysis. The table reports values of aneuploidy and heterogeneity of the karyotype as represented in Fig. 5. Cells in the hESC state show no difference between ATRX[+/+] and ATRX[−/−] genotypes. Differentiation induces an increase in both aneuploidy and heterogeneity values exclusively in ATRX[−/−] cells. RTEL1[−/−] and BLM[−/−] cells have been used as positive controls.

osteosarcoma cells were obtained from the UC Berkeley Cell Culture Facility. HeLa 1.3 and U2OS were maintained in fibroblast medium (DMEM (Gibco) supplemented with 15% FB Essence (Seradigm), 1 mM glutamine (Sigma-Aldrich), 1% non-essential amino acids (Gibco), and 100 U/mL Penicillin–Streptomycin (Gibco)) and passaged every 3–5 days enzymatically with Trypsin-EDTA (0.25%) (Gibco). Trypsin was inactivated by either wash medium or fibroblast medium.

Transient feeder-free hESC and E7-differentiated cultures were maintained in E7 medium (DMEM/F12 (Gibco) supplemented with 10% FB Essence (Seradigm), 64 mg/L L-ascorbic acid (Sigma-Aldrich), 14 µg/L sodium selenium (Sigma-Aldrich), 100 µg/L FGF-Basic (AA 1–155) (Gibco), 19.4 mg/L insulin (Sigma-Aldrich), 543 mg/L NaHCO₃ (Sigma-Aldrich), and 10.7 mg/L transferrin (Sigma-Aldrich). Cells were grown on tissue culture plates treated with Matrigel matrix (Corning) and passaged using Trypsin-EDTA (0.25%) (Gibco).

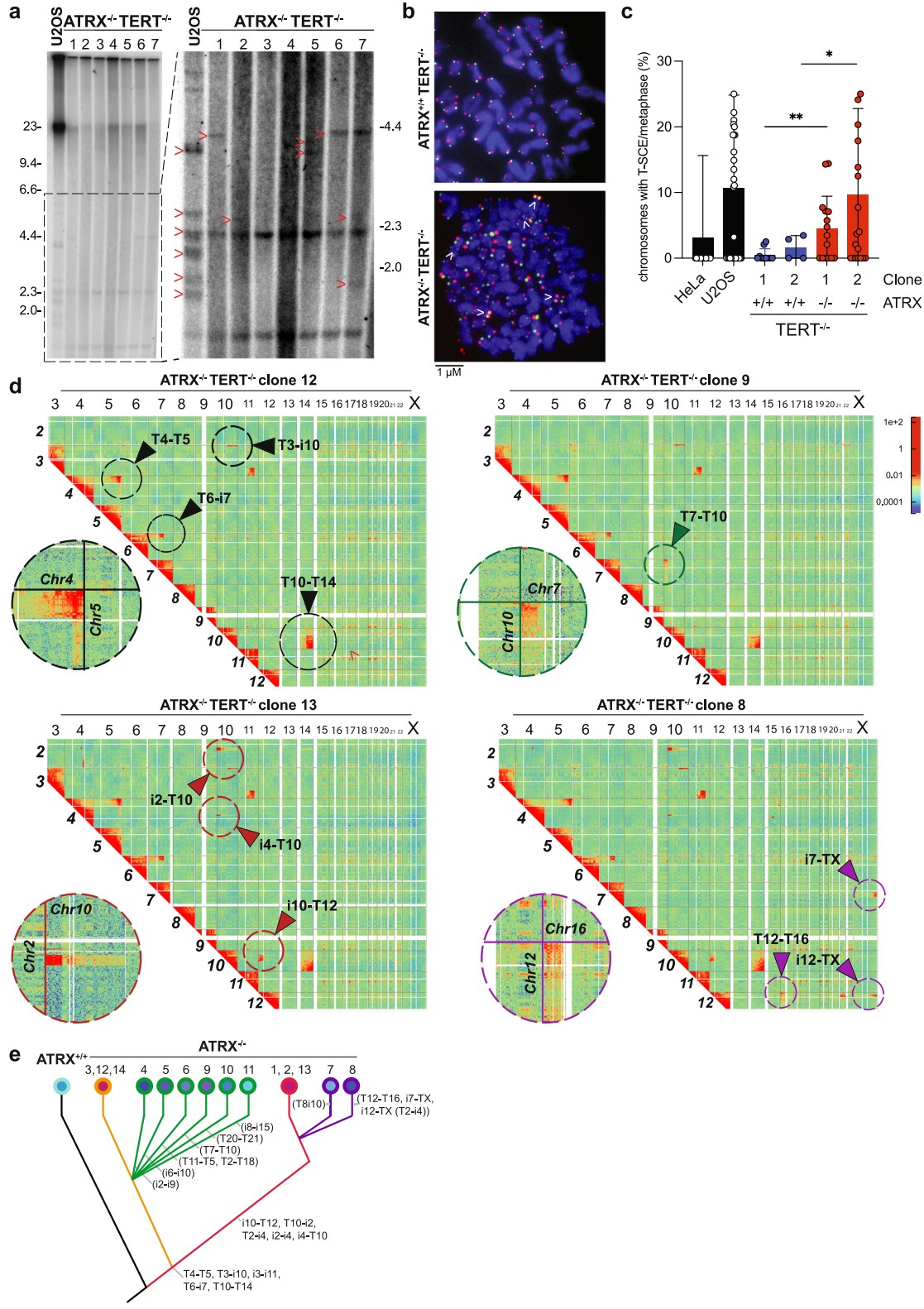

Cell culture images were acquired with a Zeiss AXIO Observer A.1 microscope using the Zeiss Zen 2012 (blue edition) 64-bit software.

All cells tested negative for *Mycoplasma* contamination monthly.

**Genome editing in hESCs**

All targeting was performed as previously described[42,55,56,65,66]. CAS9 and sgRNAs were expressed using the pX330 plasmid[67]. Each targeting step was performed by co-electroporation of $1-2 \times 10^7$ hESCs with 15 μg

of each pX330 plasmid and 7.5 μg of GFP expression plasmid. 48-72 h later cells were sorted for GFP fluorescence and single cell-derived hESC colonies were isolated and genotyped by Southern blotting or PCR followed by Sanger sequencing.

Guide sequences for spCAS9-mediated genome editing are as follows: ATRX knockout by exon 1 excision (guide 1: gctttggaggagg tagccaa; guide 2: acatgaccgctgagcccatg), BLM knockout by single guide indel generation (tctctatgagaggaagctct); RTEL1 knockout by

**Fig. 6 | Differentiated ATRX$^{-/-}$ clones show signs of telomere-driven chromosomal instability. a** Telomere restriction fragment blot of different ATRX$^{-/-}$ clones along with U2OS cells. Left panel. represents the whole lanes, right panel is a magnification highlighting telomeric fusions. DNA fragments sizes are indicated along the panels in kilobases. **b** Maximum projection images of metaphase spreads stained for CO-FISH analysis in TERT$^{-/-}$, ATRX$^{+/+}$ and ATRX$^{-/-}$ cells (see "Methods"). TelG probe is here represented in green, while TelC probe is shown in red, chromosomes are counterstained with DAPI and here shown in blue. Arrows indicate telomere sister chromatid exchanges (T-SCEs). **c** Quantification of CO-FISH assay. Telomeric sister chromatids exchanges (T-SCE) were scored for each chromosome end and normalized on the total number of chromosomes analyzed. A total of roughly 3000 chromosomes were blinded and manually scored. Two independent experiments were pooled together and are here represented. Data shown are analyzed from the total number of chromosomes in the two experiments (bars) with means ± s.d., dots represent the average value for each metaphase analyzed.

Asterisks represent $p$ value (respectively $p < 0.009$ for ATRX$^{+/+}$ clone 1 vs ATRX$^{-/-}$ clone 1 and $p < 0.019$ for ATRX$^{+/+}$ clone 2 vs ATRX$^{-/-}$ clone 2) as calculated by two-sided Welch's $t$ test. ATRX$^{+/+}$: $n = 19$ for clone 1, $n = 9$ for clone 2; ATRX$^{-/-}$: $n = 30$ for clone 1, $n = 36$ for clone 2; HeLa: $n = 27$; U2OS: $n = 44$. Source data are provided as a Source data file. **d** Micro-C chromosome maps of TERT$^{-/-}$, ATRX$^{-/-}$ clones obtained by the HiGlass software (see "Methods"). The color bar legend represents the number of reads. Dotted circles highlight contacts involving terminal chromosomal regions and magnifications of different contacts are also shown. Telomeric contacts are indicated as "T" while chromosomal internal contacts are indicated with as "i", followed by the corresponding chromosome number. **e** Cladogram representing telomere contacts in on representative ATRX$^{+/+}$ clone (black line) and different ATRX$^{-/-}$ clones (numbers indicate each clone). Reconstitution of contacts identified by micro-C analysis shows the ongoing evolution of telomere fusions in ALT cells. Several telomeres (T) and internal (i) chromosomal contacts are reported at each branch of the cladogram and define each cell clone.

single guide indel generation (tggcgagaacacctccgaga); CDKN2A knockout by exon 2 excision (guide 1: accattctgttctctctggc; guide 2: cgcggaaggtccctcaggtg).

Cre-mediated recombination was performed by co-transfection of StemMACS Cre recombinase mRNA (Miltenyi Biotec) with Stemgent eGFP mRNA (Milltenyi Biotec) into hESCs using StemFect RNA Transfection Kit (ReproCELL) according to manufacturer instructions. Twenty-four to 72 h later, cells were sorted for GFP fluorescence and single-cell-derived hESC colonies were isolated and genotyped by PCR. SV40LT was expressed using (addgene #58993) standard lentiviral production and infection protocols.

### Southern blotting and PCR genotyping
Southern blot analysis was performed as previously described[42,65]. *TP53* deletion was confirmed using an external 5' probe amplified from genomic DNA with primers (Fw: ttttcagacctatggaaact; Rev: ctgtagatgggtgaaaagag). *CDKN2A* deletion was confirmed using a probe 5' to the excision site amplified from genomic DNA with primers (Fw: ggggaaatgatgttggcttagaatcct; Rev: caatgaagtccttcgtcttggtca). ATRX deletion was confirmed using PCR primers (Fw: ggtgaatctcggctccacta; Rev: gaaaacgatgcctctttcgg). Cre-mediated loopout of TERT from *AAVS1* was confirmed using PCR primers (Fw: tgtcaaggtggatgtgacgg; Rev: gaggagctctgctcgatgac).

### Fibroblast differentiation
hESC colonies were lifted from the MEF feeder layer enzymatically with 1.5 mg/mL collagenase type IV (Gibco) and isolated by sedimentation and washing 3 times with wash medium. Colonies were suspended in fibroblast medium and grown in Ultra-Low Attachment Culture Dishes (Corning) for formation of embryoid bodies (EBs). Medium was replenished every 3 days by sedimentation and resuspension of EBs. After 9 days, EBs were transferred to tissue culture dishes to attach. Seven days later, EBs and fibroblast-like cells were passaged using Trypsin-EDTA (0.25%) (Gibco), triturated to single-cell suspension, and plated on tissue culture dishes. Cultures were maintained in fibroblast medium on plates treated with gelatin (Sigma-Aldrich) and were passaged every 5–7 days.

### E7 differentiation
hESC colonies were lifted from the MEF feeder layer enzymatically with 1.5 mg/mL collagenase type IV (Gibco) and isolated by sedimentation and washing 3 times with wash medium. Colonies were suspended in E7 medium and transferred to tissue culture dishes treated with Matrigel (Corning). After 7 days, cultures were passaged using Trypsin-EDTA (0.25%) (Gibco), triturated to single-cell suspension, and plated on Matrigel-coated tissue culture dishes. Cultures were maintained in E7 medium and were passaged every 5–7 days.

### Neural precursor cell differentiation
Single-cell dissociated hESCs were cultured on Matrigel-coated plates at a density of $5 \times 10^4$ cells/cm$^2$ and maintained in complete conditioned hESC medium until >90% confluent. A modified dual-SMAD inhibition protocol was performed to differentiate hESCs into NPCs as described previously[68,69]. Cells were passaged by dissociation with StemPro Accutase, split 1:3 every 5 days, and maintained in N2 medium (50% DMEM/F12 (Gibco) and 50% Neurobasal Medium (Gibco) supplemented with N-2 Supplement (Gibco), GlutaMAX (Gibco), 100 U/mL Penicillin–Streptomycin (Gibco), 0.2% insulin (Sigma-Aldrich), and 0.075% (w/v) bovine serum albumin (Sigma-Aldrich)).

### Immunocytochemistry (ICC) and EdU staining
Cells were plated onto glass coverslips and fixed in 4% paraformaldehyde solution in PBS (Sigma-Aldrich). Samples were permeabilized in permeabilization solution (PBS with 3% horse serum (Sigma-Aldrich) and 0.1% Triton X-100 (Sigma-Aldrich)). Primary antibody incubation was performed overnight in permeabilization solution at 4 °C. Samples were then washed 3 times in PBS, 5 min each wash. Secondary antibody incubation was performed for 1 h in permeabilization solution at 25 °C protected from light. Samples were washed 3 times in PBS, 5 min each wash, with 1 µg/mL DAPI (Sigma-Aldrich) added to the second wash. Coverslips were then mounted using ProLong Gold Antifade on glass slides and imaged.

For EdU staining, cells were plated on glass coverslips. 24 h prior to fixation, cells were treated with 10 µM RO-3306 (Sigma-Aldrich). 2 h prior to fixation, 10 µM EdU was added to cell culture media. Samples were then fixed in 4% paraformaldehyde solution in PBS. EdU detection was then performed using the Click-iT Plus EdU Alexa Fluor 647 Imaging Kit (Thermo Fisher). For further ICC steps, samples were subsequently treated as described above.

All microscopy images were acquired using a Nikon Eclipse TE2000-E epifluorescence microscope using the NIS-Element 4.51.00 64-bit software.

### C-circle assay
The C-circle assay was performed as previously described[70,71]. Briefly, extracted genomic DNA was digested in *Eco*RI (New England BioLabs), precipitated and extracted by phenol-chloroform, resuspended, and quantified by a Qubit™ 2.0 Fluorometer (Life Technologies). 20 ng of each sample was incubated with φ29 polymerase as previously described. Samples were attached to an Amersham Hybond-XL membrane (Fisher Scientific) by dot blot and probed with a $^{32}$P-end-labeled (CCCTAA)$_3$ oligonucleotide. Parallel membranes were probed with a 5'-gtaatcccagcactttgg-3' end-labeled oligonucleotide which binds to the Alu consensus sequence to normalize for genomic DNA content.

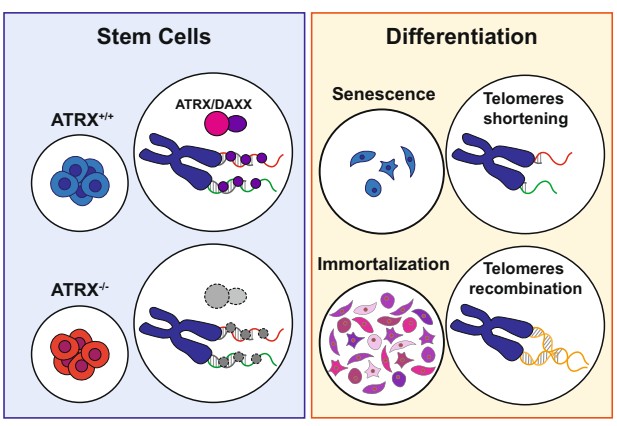

**Fig. 7 | Unscheduled telomere recombination drives ALT immortalization in ATRX$^{-/-}$ cells after differentiation.** Model of ALT immortalization. In the stem cell state, ATRX establishes that telomeres are not recombinogenic. Once differentiated, ATRX$^{+/+}$ cells repress telomere recombination, enter crisis, and senesce. ATRX$^{-/-}$ cells fail to repress telomere recombination, undergo active states of telomere-mediated genomic instability, and ultimately immortalize.

## Chromosome-orientation FISH

CO-FISH analysis was performed as previously described[72]. Briefly, 24 h prior to fixation, cells were cultured in growth medium containing 10 μM bromodeoxyuridine (BrdU) (Invitrogen). Two hours prior to fixation, 0.2 μg/mL colcemid (Roche) was added to medium. Cells were dissociated by Trypsin-EDTA (0.25%) (Gibco), centrifuged and gently resuspended for 5 min in 75 mM KCl. Cells were then centrifuged, supernatant was aspirated, and cells were gently resuspended in residual supernatant before fixation in 3:1 methanol:acetic acid. Metaphase spreads were made according to standard techniques and BrdU-containing strands were digested according to the previously described procedures[72]. Telomeres were hybridized sequentially with TelG-Alexa488 and TelC-Cy3 (PNA Bio). After dehydration, slides were stained with 1 μg/mL DAPI (Sigma-Aldrich) and mounted with ProLong Gold before imaging.

## Telomere and centromere FISH

Cells were cultured in growth medium and 50 ng/mL Nocodazole were added 16 h prior to fixation. Metaphase spreads were prepared as previously described. Telomere and centromeres were concomitantly hybridized with TelC-Cy3 (PNA Bio) and CENPB-Cy5 (PNA Bio) for 2 h at RT after denaturation at 75 °C. After dehydration, slides were stained with 1 μg/mL DAPI (KPL) and mounted with ProLong Gold before imaging.

## Image acquisition and analysis

All ICC and CO-FISH micrographs were taken on a Nikon Eclipse TE2000-E with a 100× objective using an Andor Zyla 4.2 sCMOS camera. Images were acquired using Nikon NIS-Elements. For experiments in which two or more conditions were quantitatively compared, the same exposure and acquisition settings were used for each image. APB, TIF, and EdU colocalization with telomeres were automatically processed using ImageJ with ComDet plugin.

## Strand-seq and bioinformatics

Cells were prepared for Strand-seq analysis as previously described[73]. Briefly, hESCs were collected after BrdU pulse and resuspended in nuclei staining buffer (100 mM Tris-HCl pH 7.4, 150 mM NaCl, 1 mM CaCl$_2$, 0.5 mM MgCl$_2$, 0.1% NP-40, and 2% bovine serum albumin (Sigma-Aldrich) supplemented with 10 μg/mL Hoechst 33258 (Enzo) and 10 μg/mL propidium iodide (Sigma-Aldrich)). Single nuclei were sorted into 5 μL ProFreeze-CDM freeze medium (Lonza) + 7.5%

dimethyl sulfoxide in 96-well skirted PCR plates (4titude), based on low propidium iodide (G1 phase) and low Hoechst (BrdU-induced quenching) fluorescence using an Influx cell sorter (BD Biosciences). Strand-seq libraries were prepared using an Agilent Bravo liquid handling platform as described before[73]. For each experiment, 96 libraries were pooled and 250–450 bp-sized fragments were isolated and purified. DNA quality and concentrations were assessed using the High Sensitivity dsDNA kit (Agilent) on the Agilent 2100 Bio-Analyzer and on a Qubit 2.0 Fluorometer (Life Technologies), respectively.

Single-end sequencing reads from Strand-seq libraries were generated using the HiSeq 2500 or the NextSeq 500 sequencing platform (Illumina; up to 77 bp). Indexed reads were aligned to the human reference genome (GRCh38) using Bowtie2[74]. Only non-duplicate reads with a mapping quality greater than or equal to 10 were further analyzed as described in[75] and using an AneuFinder R-based package[76] (https://github.com/ataudt/aneufinder). Strand-seq libraries were prefiltered to avoid errors arising from low-quality data. For this, we excluded libraries with less than 25 reads/Mb, >10% background reads, no template strand inheritance, uneven coverage (high number of copy number segments) or libraries from cells in second cell division. Libraries passing these quality criteria served as input for further analysis. AneuFinder was used to locate and classify template strand switch and copy number change breakpoints. In short, following GC corrections and blacklisting of artifact-prone regions, libraries were analyzed using the edivisive copy number calling algorithm with variable width bins (binsize: 100 kb; step size: 40 kb) and breakpoint refinement ($R = 10$, confint = 0.99; other settings as default). Copy numbers for both the Watson (negative) and Crick (positive) strand were called and breakpoints were defined as changes in copy number state. As BAIT and AneuFinder also detect stable chromosomal rearrangements (e.g., inversions), template switching events that occurred at the exact same locations were excluded from the analysis. Computationally localized SCE or copy number alteration events were further manually verified by visual inspection of chromosome ideograms obtained from AneuFinder and BAIT. Aneuploidy, structural and heterogeneity scores were calculated as previously described[76].

## Harlequin chromosome analysis

Cells were incubated in 10 μM BrdU for 24 h, arrested in colcemid and metaphase spreads prepared as previously described. Slides were rehydrated in 2× SSC for 5 min, stained with Hoechst 33258 and UV-crosslinked. Sister chromatids were stained with 5% Giemsa dye in 2× SSC for 20 min, washed with water and left to dry. Images were taken with a light microscope and chromatid exchanges were manually annotated for each chromosome.

## Telomere length assessment

To collect genomic DNA, hESC lines were enzymatically released from the MEF feeder layer by treatment with 1.5 mg/mL collagenase type IV and washed 3 times in wash medium and gravitational sedimentation to minimize contaminating MEF cells. Genomic DNA was prepared as described previously[77]. MEF telomeres are resolved by size from hESC telomeres and do not interfere with analysis of telomere length. Genomic DNA was digested with *Mbo*I and *Alu*I overnight at 37 °C. Digested DNA was normalized and run on a 0.75% Seakem ME Agarose (Lonza) gel and dried under vacuum for 2 h at 50 °C. The dry gel was denatured in 0.5 M NaOH, 1.5 M NaCl for 30 min at 25 °C, then neutralized with 1 M Tris-HCl pH 6.0, 2.5 M NaCl, 2x for 15 min. The gel was then pre-hybridized in Church's buffer (1% BSA, 1 mM EDTA, 0.5 M NaPO$_4$, 7% SDS, pH 7.2) for 1 h at 55 °C before adding $^{32}$P-end-labeled (CCCTAA)$_3$ probe. The gel was washed in 4× SSC buffer 3 times for 15 min at 50 °C and once in 4× SSC + 0.1% SDS at 25 °C before exposing on a phosphorimager screen.

## Single telomere length analysis (STELA)

Single telomere length analysis (STELA) was performed as previously described[78]. hESC colonies were separated from the MEF layer by treatment with 1.5 mg/mL collagenase type IV and washed 3× in wash medium, collecting by sedimentation to minimize contaminating MEF cells. DNA was extracted from cell pellets using the Norgen Cells and Tissue DNA Isolation Micro Kit. DNA was solubilized by digestion with *Eco*RI and quantified on a Qubit 2.0 Fluorometer, then diluted to 10 ng/μL in 10 mM Tris-HCl (pH 7.5). DNA was ligated at 35 °C for 12 h in a volume of 10 μL containing 10 ng genomic DNA, 0.9 μM telorette linker, and 0.5 U T4 DNA ligase (New England Biolabs). Ligated DNA was diluted to 250 pg/μL in water and multiple PCRs were performed in volumes of 15 μL containing 200 pg ligated DNA, 0.25 μM XpYpE2 + G and teltail primers, 0.3 mM dNTPs, 7.4 mM MgCl$_2$, 1× Taq Buffer with (NH$_4$)$_2$SO$_4$, and 1 U of a 10:1 mix of *Taq* (New England Biolabs) and *Pwo* (Sigma-Aldrich) polymerase. Reactions were cycled on a Bio-Rad C1000 Touch Thermal Cycler: 25 cycles of 94 °C for 15 s, 65 °C for 30 s, 68 °C for 10 min. DNA fragments were resolved on a 0.5% agarose gel and detected by Southern blot with a random-primed α-$^{32}$P-labeled XpYp probe generated by PCR using primers (XpYpE2: ttgtctcagggtcctagtg; XpYpB2: tctgaaagtggacctatcag). Telomere lengths were estimated using TeSLA-QUANT software[79].

## Immunoblotting

Cells were collected in RIPA buffer (150 mM NaCl, 1% Triton X-100, 0.5% sodium deoxycholate, 0.1% SDS, 50 mM Tris pH 8.0) with 1 mM phenylmethanesulfonyl fluoride and cOmplete ULTRA protease inhibitor (Roche) and Halt Phosphatase inhibitor (Thermo Scientific). Protein concentration was determined by Bio-Rad Protein Assay colorimetric dye quantified by a Bio-Rad xMark microplate reader. In all, 15–20 μg protein in Laemmli sample buffer was loaded onto 5% (ATRX) or 10% (DDR proteins) polyacrylamide gels. Proteins were transferred to nitrocellulose membranes (Bio-Rad), blocked in 5% BSA in tris-buffered saline (TBS)-Tween 20 for 1 h at 25 °C, then incubated with primary antibodies diluted in 5% BSA in TBS-T overnight at 4 °C. Membranes were then washed 3 × 15 min in TBS-T and incubated in horseradish peroxidase-conjugated secondary antibodies (Bio-Rad) for 1 h at 25 °C, washed, incubated with Clarity Western ECL substrate (Bio-Rad) before imaging on a Bio-Rad ChemiDoc XRS+. Membranes were stripped by 2 × 10 min incubation at 25 °C in stripping buffer (200 mM glycine, 0.1% SDS, 1% Tween 20, pH 2.2) before re-blocking and incubation with subsequent primary antibodies.

A list of antibodies and respective dilutions can be found in the Supplementary Table 7.

## qRT-PCR analysis

RNA was isolated using Trizol (Invitrogen) extraction followed by ethanol precipitation. Reverse transcription was performed on 1000 ng of total RNA by oligo(dT) and random priming using the iScript cDNA Synthesis Kit (Bio-Rad). qRT-PCR was performed in a CFX96 (Bio-Rad) with KAPA SYBR FAST master mix ROX low (Roche).

## Telomerase catalytic activity assay

PCR-based telomeric repeat amplification protocol (TRAP) was performed as previously described in ref. [80]. Primers TS (aatccgtcgagcagagtt) and ACX (gcgcggcttaccccttacccttacccttaccctaacc) were used for amplification of telomeric repeats. TSNT (aatccgtcgagcagagttaaaaggccgagaagcgat) and NT (atcgcttctcggcctttt) were used as an internal control. Protein extracts were generated by repeated freeze–thaw cycles in hypotonic lysis buffer (HLB) (20 mM HEPES, 2 mM MgCl$_2$, 0.2 mM EGTA, 10% glycerol, 1 mM dithiothreitol, 0.1 mM PMSF, 0.5% CHAPS). Protein concentrations were determined by Bio-Rad Protein Assay colorimetric dye quantified by a Bio-Rad xMark microplate reader. 200 ng of total protein were used for input into $^{32}$P-dGTP PCR. TRAP products were resolved on a 10% polyacrylamide

in 1× TAE gel. Dried gels were visualized by exposure on a phosphorimager screen.

## hESC colony cell counting

To measure hESC population doubling, hESC colonies were grown feeder-free in E7 medium supplemented with ROCK inhibitor (Y-27632) (Chemdea) for 24 h, then treated with Trypsin-EDTA (0.25%) for single-cell suspension. hESCs were plated at low density (1000 cells/ 10 cm plate) on tissue culture plates coated with Matrigel. After 72 h, cells were washed with PBS and fixed with 4% paraformaldehyde in PBS. Nuclei were counterstained with 1 μg/mL DAPI. Distinct clonal colonies were imaged and nuclei counted. Population doublings were calculated assuming colonies were founded by single cells.

## Micro-C library preparation and analysis

Cells for each genotype were subcloned by serial dilution and isolated colonies were picked and expanded. Micro-C assay was performed following Dovetail Micro-C kit protocol on 1 × 10$^6$ cells. Briefly, frozen cells were resuspended in 1× PBS and crosslinked with 3 mM DSG and 1% formaldehyde. Washed cells were digested with 0.5 μL MNase in 100 μl of Nuclease digest buffer with MgCl$_2$. Digestion was stop by addition of 5 μL of EDTA 0.5 M. Chromatin was solubilized by addition of 3 μL of 20% SDS and captured on chromatin capture beads. Chromatin bound DNA ends were repaired and A-tailed using the End Polishing enzyme mix and buffer. Dovetail bridge was then ligated to the chromatin bound DNA free ends following the Micro-C kit protocol then subjected to intra-aggregate ligation to create chromatin–chromatin long range interaction. Finally, the DNA bound to chromatin was recovered by proteinase K treatment and crosslink reversal and purified using SPRIselect beads (Beckman Coulter). Following DNA quantification by Qubit, the DNA was converted to an Illumina compatible library following the Dovetail library module (Dovetail Genomics). Libraries were sequenced on an Illumina HiSeq 4000. Raw fastq files were aligned using BWA mem version 0.7.17-r1198-dirty with the −5SP options with an index containing only the 24 main chromosome from the human genome release hg38 (available from the UCSC genome). The aligned paired reads were annotated with pairtools parse (https://github.com/open2c/pairtools) with the following options−min-mapq 40−walks-policy 5unique−max-inter-align-gap 30 and the−chroms-path file corresponding to the size of the chromosome used for the alignment index. The paired reads were further processed to remove duplicated reads, sorted with unaligned reads removed with the pairtools sort and the pairtools dedup tools with the basic option to produce an alignment file in the bam format as well as the location of the valid pair. The valid pairs were finally converted to the.cool and.mcool format using the cooler cload and cooler zoomify tools[81] and to the.hic format using the juicer tool[82]. Chromosomal maps were obtained using HiGlass viewer tool[83]. All tracks were set at the same threshold and telomeric interactions were manually scored by comparison of the ATRX$^{−/−}$ with the ATRX$^{+/+}$ contact maps.

## Statistics and reproducibility

Statistical details including statistical tests, values of $n$, significance definitions, and dispersion measures of experiments can be found in the relevant figure legends. Sample sizes were chosen based on previous studies. All samples were included in the analysis. Quantifications were performed by software, but investigator was not blind to genotype. Unless specifically specified otherwise, all experiments were performed once under the experimental conditions reported in this work. Statistical analysis was performed using GraphPad Prism 8 software for Windows and the specific test for each experiment is noted in the appropriate figure legend. An unpaired two-tailed $t$ test was used to compare the means of two normally distributed groups. A one-way analysis of variance (ANOVA) was used to compare the means of three or more groups, followed by Tukey's multiple comparisons test. For

sets with three or more groups with non-normal distributions, a Kruskal–Wallis test was used followed by Dunn's multiple comparisons test.

## Reporting summary

Further information on research design is available in the Nature Portfolio Reporting Summary linked to this article.

## Data availability

Micro-C next-generation sequencing data generated in this study have been deposited in NCBI's Gene Expression Omnibus and are accessible through GEO Series accession number GSE212809. Strand-seq next-generation sequencing data generated in this study have been deposited in the ArrayExpress database (http://www.ebi.ac.uk/arrayexpress) under accession code E-MTAB-12582. Source data are provided with this paper.

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

## Acknowledgements

We thank all members of the Hockemeyer lab for advice and critical comments on the manuscript. We thank Rebecca Bartke, Fernando Rodriguez Perez, and Dan Kramer for help with experiments at the early stages of the project. D.H. is a Chan Zuckerberg Biohub Investigator and supported by a Research Scholar Grants from the American Cancer Society (133396-RSG-19-029-01-DMC). D.H. is a Pew-Stewart Scholar for Cancer Research supported by the Pew Charitable Trusts and the Alexander and Margaret Stewart Trust. The work in the Hockemeyer laboratory was supported by the Siebel Stem Cell Institute, the Chan Zuckerberg Biohub, and D.O.D. (W81XWH-19-1-0586) and a grant by the Laboratory of Genomics Research (LGR).

## Author contributions

T.K.T., A.M., J.J.S., S.G.R. and D.H. designed and conducted the human stem cell experiments. D.C.J.S. and P.M.L. conducted the strand-sequencing experiments. M. Bhakta performed the Micro-C sequencing experiments and M. Blanchette, A.M. and D.H. analyzed the Mirco-C data. T.K.T., A.M. and D.H. wrote the manuscript with input from all authors.

## Competing interests

M. Bhakta and. M. Blanchette are full-time employees of Dovetail Genomics. The remaining authors declare no competing interests.
