## [Peer Review File · Nature Communications]

A non-genetic switch triggers alternative telomere lengthening and cellular immortalization in ATRX deficient cellsREVIEWER COMMENTS

Reviewer #1 (Remarks to the Author):

Turkalo et al

An epigenetic switch triggers alternative telomere lengthening and cellular immortalization in ATRX deficient cells

This is an interesting report which casts new light on how the alternative lengthening of telomere (ALT) pathway is triggered in cancer cells.

They make a number of observations:

1. In an ATRX knockout (KO) model in human embryonic stem cells (hESCs) once differentiated into fibroblasts when checkpoint proficiency is prevented by SV40 these fibroblasts become immortalised and display features of ALT.
2. ATRX KO hESCs rapidly differentiated into fibroblasts (by E7 protocol) exhibit ALT features in the absence of SV40 and without becoming immortalised showing these are not prerequisites for manifesting features of ALT. KO of p53 in these cells is sufficient to immortalise these cells.
3. Differentiation of ATRX KO hESCs which lack p53 and p16 into fibroblasts or neural progenitor cells exhibit ALT features irrespective of the expression of telomerase suggesting telomeric crisis is not a prerequisite for triggering ALT but rather the epigenetic state of the cells is.
4. TERT KO/ATRX KO fibroblasts achieve immortalisation whereas TERT ko/ATRX wildtype fibroblasts cease to proliferate.
5. Differentiation of ATRX KO cells leads to increased genomic instability.

Specific comments:

There is a lot of sloppiness suggesting a rushed submission.

The numbering of the figures is very confusing and there is reference to extended data and data figures preceded by S. Some data referred is missing:

Where is extended data Fig 1J and data in the text "Analysis for 5 independent ATRX^{-/-}, TERT^{-/-} cell lines and 3 ATRX^{+/+}, TERT^{-/-} revealed that the proliferative capacity of cells after TERT loop out correlated with the starting telomere length" referred to as extended data Fig 3A.

There is no explanation in Extended data Fig 3F legend to what is ATRXch

Contrary to the text, Figure 3G does not compare ATRX KO with ATRX WT cells.

Extended data Fig 3H shows effects of ATRX KO not TERT KO contrary to the text.

Referring to extended data Fig 3J, the text says "Single telomere length analysis (STELA) confirmed that ATRX^{+/+}, TERT^{-/-} and ATRX^{-/-}, TERT^{-/-} hESCs stopped proliferating at a similar telomere length despite their different proliferative capacity" but it is not clear from this data that the cells have stopped proliferating.

I don't think the data presented is sufficient to support this statement "This indicated that ATRX loss-of-function did not result in spontaneous loss of telomeric sequence and premature crisis. Our data shows that telomere crisis is insufficient to trigger ALT and cellular immortalization of ATRX^{-/-} hESCs."

"Importantly, for ATRX^{-/-} TERT^{c/c} cells, but not ATRX^{+/+} TERT^{c/c} cells, differentiation into NPCs or fibroblast like cells resulted in establishment of long and heterogeneous telomeres compared to contemporaneous ESC cultures (Extended data Fig. 5D). We conclude that the epigenetic status is responsible for ALT phenotypes and cellular immortalization in ATRX^{-/-} cells." At this stage of the report the authors haven't demonstrated immortalisation only long telomeres.

“However, ATRX^{-/-}, TERT^{-/-} cultures recovered and continued to proliferate after their contemporaneous ATRX^{-/-}, TERT^{-/-} hESC cultures went through crisis and died.” No data is shown to support this statement for hESCs.

“A surprising result is that telomeres in differentiated ATRX^{-/-}, TERT^{-/-} cells that proliferated for more than 250 days had a similar length and heterogeneity as those shortly after differentiation. This suggests that additional factors contribute to the proliferation and eventual immortalization of these cells rather than simply telomere length.” The change in proliferation rate at around 150 days post TERT KO in Fig 4E is more suggestive of this. You can’t tell about the potential for immortalisation from data in Fig 4G.

“So far, our data showed that loss of ATRX in a permissive epigenetic environment renders telomeres as recombinogenic sites of DNA damage, independent of telomere length”. I don’t understand how they argue that this is independent of telomere length. They have shown that loss of ATRX in diffd cells leads to recombination at telomeres independent of telomerase but not telomere length.

There is a discrepancy in the data of TSEs/recombination in differentiated ATRX KO cells as assayed by Strand-seq vs Micro-C which needs explaining. How do you know that Micro-C is showing genomic rearrangements rather than chromatin interactions?

In Fig 6B the authors need to show an example of telomere fusion. Also only 3/5 of the arrowed examples look like SCE.

In Fig 6d need to explain in legend the difference between T and i.

The authors might like to comment on the contrast between their findings regarding ATRX depletion in human ES cells and what is seen in mouse ES cells:

Wong LH et al 2010 Genome Research - Strong TIFs in mES cells. Also on differentiation both ATRX and H3.3 were less associated with telomeres suggesting less requirement for ATRX in differentiated cells.

Reviewer #2 (Remarks to the Author):

In this manuscript the Hockemeyer group focuses on the requirements for ALT activation, using a novel and unique system, where gene targeting in human pluripotent stem cells is employed to test ALT requirements upon stem cell differentiation. ATRX deficient p16 negative hESC were differentiated into fibroblasts, whereupon transduction with SV40LT allowed about 1% of the cells to survive and form clones that exhibited all recognized signs of ALT. Alt induction was dependent on ATRX, but continued proliferation required p53 suppression in addition. Differentiation was a required step in the ALT activation program, since p16/p53/ATRX negative hESC did not acquire ALT phenotypes when telomere crisis was initiated as consequence of telomerase deletion. Remarkably, telomerase expression was fully compatible with ALT activation upon differentiation. Differentiation led to the establishment of immortality via ALT, since removal of telomerase did not impact long-term proliferation. Sophisticated genome analysis revealed that ATRX loss does not lead to major genome instability, but ongoing ALT subsequently does. Instability was instigated by telomere fusions, which were highlighted by Micro-C. In conclusion, the authors suggest a novel model, where ATRX loss generates an epigenetic environment conducive of ALT activation.

In summary, this is a thorough and novel study evaluating ALT activation, using an innovative and powerful model. Several discoveries render the approach here of high interest to a wide readership:

- AKT activation is independent of telomere crisis
- P53 loss is required for ALT dependent long-term growth
- Telomerase expression is not incompatible with ALT activation

- Accumulation of genome instability is initiated by end-to-end fusion

As such, the manuscript is undoubtedly suitable for Nature Communications and will be a highly cited novel contribution to the field.

A few points should be addressed in a revision. I understand that some of these are complicated and might exceed the scope of the manuscript:

- 1) What does the loss of CDKN2A/p16 do in the early stages of ALT activation? If I understand this correctly (authors, please indicate if I am wrong), ATRX deletion in hESC does not lead to viable cells unless p16 is co-deleted. Is it possible that p16 loss contributes to the epigenetic environment that allows ATRX to establish ALT predisposition? Similarly, are the hESC in Fig2 also p16 deficient?
- 2) What happens to the 99% of cells that don't survive in the Fig 1 experiment? I understand that this might be hard to impossible to evaluate experimentally, but it would be interesting to read the authors' view here. Is it possible that the cell death resembles a crisis event?
- 3) Is it possible to add stats/error bars to the c-circle experiments?
- 4) Many established ALT lines have lost cGAS/STING signaling, presumable to prevent the extrachromosomal DNA to activate innate immune signaling. Is this pathway still active in the immortal ALT cells here? Is there a high degree of phosphor TBK signaling?
- 5) Is it possible to tell whether telomerase is active in the ALT cells where TEERT has not been deleted? In other words, is it possible to tell whether recombination exists in parallel with telomerase-based telomeric nucleotide incorporation? Similarly, do the authors know whether residual TERT activity in ATRX negative ALT positive cells prevents genome instability accumulation, presumably by preventing some of the initial end to end fusions?
- 6) Please remove the word 'significant' from page 8, second paragraph, last sentence, unless backed up by stats.

Reviewer #3 (Remarks to the Author):

Turkalo et al present evidence that induction of ALT is modulated not by telomere status (length or damage), but rather by cellular state. The authors utilize the differentiation of human embryonic stem cells to reach this conclusion, showing that despite ATRX loss (in different genetic backgrounds and different telomere lengths), hESCs do not show several molecular characteristics usually associated with ALT. This immediately changes when the authors differentiate these cells to fibroblasts (and neural progenitor cells), where, regardless of telomere damage, loss of ATRX promotes these molecular markers of ALT.

While I agree this phenotype is interesting, the authors keep claiming (including in their title), that an epigenetic switch controls this change. The authors actually claim that they "demonstrate that an epigenetic switch occurs in ATRX-negative stem cells upon differentiation that allows for telomere recombination and cellular immortalization". No evidence at all is presented for this. No experimental support is presented for such a bold suggestion. No data presenting any sort of epigenetic analysis is shown. This significantly hampers enthusiasm with this manuscript.

Technically, the manuscript could be improved, as p-values are missing, figure legends are poorly described and the amount of data presented (sometimes repetitive) makes this an extremely challenging manuscript to read. The quality of images for some of the IFs (see below) and the light microscopy (1A and 4A) should be improved.

Additionally, the authors make several bold claims on how their data correlates with what is observed with ALT in tumors, again without presenting any evidence for that. This makes the text unnecessarily long and at times reading more like a review than a primary manuscript (this is particularly the case in the Discussion).

Some more specific comments:

Figure 1:

- Legend says "Only ATRX^{-/-} proliferated after SV40 transduction". But this is not the case, as ATRX^{+/+} fibroblasts normally proliferate (Sup. Figure 1E). Do the authors mean: ATRX^{-/-} fibroblasts only proliferate after SV40 transduction?
- Authors should show that ATRX^{+/+} fibroblasts (with and without SV40) do not show APBs
- TRF experiments are not convincing to show ALT (heterogeneity could have come from differentiation). A negative control (ATRX^{+/+}) should be added (different fibroblast clones).
- Figures 1D-G: why are authors comparing with HeLa, if they have an isogenic, fibroblast control (ATRX^{+/+})? These cells would be a much better control. (HeLa cells could be kept of course, as these are non-ALT).
- Figure 1F: what is the additional "neg" sample shown specifically on that figure (not described in the legend)

Figure 2:

- Images shown in 2A are not convincing. ATRX^{-/-} fibroblasts only have 2 TRF1 foci (this should be specifically quantified), and several foci in ATRX^{-/-} hESC seem to colocalize. More representative images should be shown.
- Figure 2F: telomeres seem heterogeneous and elongated also in ATRX^{-/-} hESC, not only ATRX^{-/-} fibroblasts.
- Figure 2D: c-circle images should be shown, not only quantification. And again, it does seem like ATRX^{-/-} hESC have an increase? Also, how many times have c-DNA analysis been performed?
- Is p53 activated in ATRX^{-/-} fibroblasts and hESCs? This should be shown. Also, authors should show comet assays for DNA damage analysis in both cell types and both genotypes.

Figure 3:

- Figures 3 and Sup. 3 are interesting but extremely confusing. Some of it is understandable, as the authors show many different genotypes in different cellular states. But some of it could be made better, which would facilitate data interpretation. Figure 3B is sub-par. C-circles should be incorporated to main figure, and again, they seem to show that hESCs do show c-circles upon ATRX deletion? Authors should compare side-by-side (hESC vs fibroblast, different genotypes)
- SF3H: why would the deletion of TERT immediately cause a slow in proliferation in ATRX^{-/-} hESCs and not in ATRX positive hESCs?

Figure 5:

- Supplemental Table 4 should be added to Figure 5 for clarity.

Figure 6:

- U2OS cells should be added as control to 6B-C
- On SF6H, the difference on SCEs is impossible to visualize. 1) Why? Shouldn't the ATRX^{-/-} cells have a significantly higher number of SCEs? 2) Despite this minimal difference, the authors show significance. The graph should be "broken" in two, at different scales, for differences to be visible.

Finally:

- It would be interesting to perform the opposite experiment: if a panel of ALT positive cells is reprogrammed back to a pluripotent state, is ALT silenced upon induction of pluripotency?

REVIEWER COMMENTS

We thank the reviewers for their positive and constructive evaluation of our work. Below we provide a detailed list of the changes and additional experiments we now include to address the reviewers points.

Reviewer #1 (Remarks to the Author):

Turkalo et al

An epigenetic switch triggers alternative telomere lengthening and cellular immortalization in ATRX deficient cells

This is an interesting report which casts new light on how the alternative lengthening of telomere (ALT) pathway is triggered in cancer cells.

They make a number of observations:

1. In an ATRX knockout (KO) model in human embryonic stem cells (hESCs) once differentiated into fibroblasts when checkpoint proficiency is prevented by SV40 these fibroblasts become immortalised and display features of ALT.
2. ATRX KO hESCs rapidly differentiated into fibroblasts (by E7 protocol) exhibit ALT features in the absence of SV40 and without becoming immortalised showing these are not prerequisites for manifesting features of ALT. KO of p53 in these cells is sufficient to immortalise these cells.
3. Differentiation of ATRX KO hESCs which lack p53 and p16 into fibroblasts or neural progenitor cells exhibit ALT features irrespective of the expression of telomerase suggesting telomeric crisis is not a prerequisite for triggering ALT but rather the epigenetic state of the cells is.
4. TERT KO/ATRX KO fibroblasts achieve immortalisation whereas TERT ko/ATRX wildtype fibroblasts cease to proliferate.
5. Differentiation of ATRX KO cells leads to increased genomic instability.

Specific comments:

There is a lot of sloppiness suggesting a rushed submission.

The numbering of the figures is very confusing and there is reference to extended data and data figures preceded by S. Some data referred is missing:

Where is extended data Fig 1J and data in the text “Analysis for 5 independent ATRX^{-/-}, TERT^{-/-} cell lines and 3 ATRX^{+/+}, TERT^{-/-} revealed that the proliferative capacity of cells after TERT loop out correlated with the starting telomere length” referred to as extended data Fig 3A.

We would like to thank the reviewer for catching this mistake; we now call out the correct figures. We provide a corrected description of the figure panel in addition to a collective summary of the experiments presented in this paragraph. The text now reads: “Analysis for 5 independent ATRX^{-/-}, TERT^{-/-} and 3 ATRX^{+/+}, TERT^{-/-} cell lines revealed that the proliferative capacity of cells after TERT loop out correlated with the starting telomere length (Fig. 3I and Supplementary 3I). Single telomere length analysis (STELA) confirmed that ATRX^{+/+}, TERT^{-/-} and ATRX^{-/-}, TERT^{-/-} hESCs stopped proliferating at a similar telomere length despite their different proliferative capacity (Supplementary Fig. 3J). Collectively, these experiments show that continuously passaging the same number of hESCs (approximately 5-10*10⁶) into telomere crisis does not lead to the

recovery of ALT positive immortalized hESC clones. TRF and STELA analysis of DNA samples collected shortly before the cultures stopped proliferating suggest that telomeres in both ATRX^{+/+} and ATRX^{-/-} cells are critically short as previously documented in TERT^{-/-} hESCs lines (Supplementary Figure 3J).”

There is no explanation in Extended data Fig 3F legend to what is ATRXch

We apologize for this oversight. We have corrected the figure legends and have assigned the denomination of “compound heterozygous”, as these clones are derived from different editing outcomes at the CAS9 cut site in the two ATRX knockout alleles. Part of the panel in previous Supplementary Figure 3F has now been moved to the main text as Figure 3G. The figure legend now reads: “(G) C-circle assay of different ATRX^{+/+}, ATRX^{ch} (compound heterozygous) and ATRX^{-/-} clones. SaOS and U2OS cells are used as positive controls in the assay. Signal was quantified and normalized on Alu probe and SaOS signal (see Supplementary Fig. 3G).” The control blot is now presented as Supplementary Figure 3G and the legend has been reworded as “(G) Alu control membrane for C-circle assay of different ATRX^{+/+}, ATRX^{ch} (compound heterozygous) or ATRX^{-/-} clones together with ATRX^{+/+} cell lines, SaOS2 and U2OS.”

Contrary to the text, Figure 3G does not compare ATRX KO with ATRX WT cells.

Yes, the reviewer is correct. The panel is now presented as Figure 3H. The text now reads “Telomeres in the ATRX^{-/-}, TERT^{-/-} hESCs shortened as expected when compared to the parental ATRX^{-/-}, TERT^{c/c} cells (Fig. 3H)”.

Extended data Fig 3H shows effects of ATRX KO not TERT KO contrary to the text.

We would like to thank the review for pointing this out. This panel is now presented as Supplementary Figure 3I. The text has been changed to: “ATRX^{-/-} hESCs showed a slower proliferation after Cre-mediated loopout of TERT from the AAVS1 locus (Supplementary Fig. 3I).”

Referring to extended data Fig 3J, the text says “Single telomere length analysis (STELA) confirmed that ATRX^{+/+}, TERT^{-/-} and ATRX^{-/-}, TERT^{-/-} hESCs stopped proliferating at a similar telomere length despite their different proliferative capacity” but it is not clear from this data that the cells have stopped proliferating.

Yes, we agree with the reviewer that this needs to be phrased more precisely. The text now reads “Collectively, these experiments show that continuously passaging the same number of hESCs (approximately 5-10*10⁶) into telomere crisis does not lead to the recovery of ALT positive immortalized hESC clones. TRF and STELA analysis of DNA samples collected shortly before the cultures stopped proliferating suggest that telomeres in both ATRX^{+/+} and ATRX^{-/-} cells are critically short as previously documented in TERT^{-/-} hESCs lines (Supplementary Figure 3J).”

I don't think the data presented is sufficient to support this statement “This indicated that ATRX loss-of-function did not result in spontaneous loss of telomeric sequence and premature crisis. Our data shows that telomere crisis is insufficient to trigger ALT and cellular immortalization of ATRX^{-/-} hESCs.”

We thank the reviewer for pointing this out. We have corrected this together with the statement above to be more precise. As we indicate in the above comment, this conclusion now reads: “Collectively, these experiments show that continuously passaging the same number of hESCs (approximately 5-10*10⁶) into telomere crisis does not lead to the recovery of ALT positive

immortalized hESC clones. TRF and STELA analysis of DNA samples collected shortly before the cultures stopped proliferating suggest that telomeres in both ATRX^{+/+} and ATRX^{-/-} cells are critically short as previously documented in TERT^{-/-} hESCs lines (Supplementary Figure 3J)."

"Importantly, for ATRX^{-/-} TERT^{c/c} cells, but not ATRX^{+/+} TERT^{c/c} cells, differentiation into NPCs or fibroblast like cells resulted in establishment of long and heterogeneous telomeres compared to contemporaneous ESC cultures (Extended data Fig. 5D). We conclude that the epigenetic status is responsible for ALT phenotypes and cellular immortalization in ATRX^{-/-} cells." At this stage of the report the authors haven't demonstrated immortalisation only long telomeres.

We apologize for this mistake. We have now changed the text to: "We conclude that differentiation, and therefore the change of epigenetic status in ATRX^{-/-} cells, is responsible for the rapid appearance of the ALT phenotypes."

"However, ATRX^{-/-}, TERT^{-/-} cultures recovered and continued to proliferate after their contemporaneous ATRX^{-/-}, TERT^{-/-} hESC cultures went through crisis and died." No data is shown to support this statement for hESCs.

We apologize for this mistake. We did not want to state that the hESCs went through crisis. The text now reads "Of note, differentiated ATRX^{-/-}, TERT^{c/c} had long and heterogeneous telomeres, had telomerase activity and could continuously be maintained in culture (Supplementary Figure 4J-K). ATRX^{+/+}, TERT^{-/-} cells failed to immortalize when continuously passaged as undifferentiated hESCs and when differentiated. In contrast, ATRX^{-/-}, TERT^{-/-} differentiated cultures recovered and continued to proliferate."

"A surprising result is that telomeres in differentiated ATRX^{-/-}, TERT^{-/-} cells that proliferated for more than 250 days had a similar length and heterogeneity as those shortly after differentiation. This suggests that additional factors contribute to the proliferation and eventual immortalization of these cells rather than simply telomere length." The change in proliferation rate at around 150 days post TERT KO in Fig 4E is more suggestive of this. You can't tell about the potential for immortalisation from data in Fig 4G.

We thank the reviewer for the comment and now the text has been changed to reference the correct figure according to the phenotype described. The text now reads: "Interestingly, differentiated ATRX^{-/-}, TERT^{-/-} cells slowed their proliferation around 150 days but then eventually kept dividing for more than 250 days (Fig. 4E). This recovery in proliferation rate cannot be attributed merely to telomeres length changes, as telomeres in ATRX^{-/-}, TERT^{-/-} cells becomes heterogeneous in length shortly after differentiation and remained largely stable over time with a significant fraction of telomere being very short (Fig. 4F-G). This suggests that additional factors, other than telomere length, contribute to the efficient immortalization of these cells."

"So far, our data showed that loss of ATRX in a permissive epigenetic environment renders telomeres as recombinogenic sites of DNA damage, independent of telomere length". I don't understand how they argue that this is independent of telomere length. They have shown that loss of ATRX in differentiated cells leads to recombination at telomeres independent of telomerase but not telomere length.

There is a discrepancy in the data of TSEs/recombination in differentiated ATRX KO cells as assayed by Strand-seq vs Micro-C which needs explaining. How do you know that Micro-C is showing genomic rearrangements rather than chromatin interactions?

We apologize that we did not explain this properly. Due to the repetitive nature of the telomeric repeats, strand-seq cannot detect T-SCE, but only SCE. The strand-seq experiment provides two insights: (1) in differentiated ATRX^{-/-} cells the genomes are unstable, as cells acquire many non-clonal alterations. (2) SCEs in ATRX^{-/-} cells, in comparison to BLM^{-/-} cells, are only marginally increased upon differentiation. The micro-C experiment does not detect T-SCE either. We use the micro-C data to map and follow telomere fusion events that we were also able to detect by conventional techniques (TRF and telomere FISH) more precisely. The data shows that cells acquire telomere/telomere fusions over time as they evolve, suggesting the presence of continuous telomere fusion events in these cells.

Although we cannot exclude the presence of transient telomeric interactions, the stereotypic topology seen for individual telomere-telomere Micro-C contacts is a strong indication of the presence of covalent telomere-telomere fusions rather than telomere associations. These types of interactions imply a preferential topology for these events that are linearly arranged and the contact frequency between DNA segments decreases as a function of the distance of the fusion point (in this case the telomere) which gives rise to the butterfly shape of the fusion in the micro-C contact map. We would expect a different pattern for a transient association or recombination intermediate. Moreover, for a telomere-telomere association to result in any signal in a contact map, two specific telomeres would need to be associated continuously in all cells of a clone. Of particular interest on the topic are the references that we now provide in the main text:

Harewood, L., Kishore, K., Eldridge, M.D., Wingett, S., Pearson, D., Schoenfelder, S., Collins, V.P. and Fraser, P., 2017. Hi-C as a tool for precise detection and characterisation of chromosomal rearrangements and copy number variation in human tumours. *Genome biology*, 18(1), pp.1-11.

Wang, S., Lee, S., Chu, C., Jain, D., Kerpedjiev, P., Nelson, G.M., Walsh, J.M., Alver, B.H. and Park, P.J., 2020. HiNT: a computational method for detecting copy number variations and translocations from Hi-C data. *Genome biology*, 21(1), pp.1-15.

The Hi-C data has now been uploaded to GEO and accession numbers for the data set are available.

In Fig 6B the authors need to show an example of telomere fusion. Also only 3/5 of the arrowed examples look like SCE.

We apologize for the labeling mistake as the arrows were misplaced. This has been corrected now in Figure 6B. In addition, we now have included the same image with only the C-strand probe in Supplementary Figure 7E, as this image demonstrates the presence of T-SCE more clearly.

In Fig 6d need to explain in legend the difference between T and i.

We thank the reviewer for finding this missing detail in our figure legends. The legends are now rewritten and include a better explanation of the telomere contacts that we identify by micro-C. The figure legend now reads "Telomeric contacts are indicated as "T" while chromosomal internal contacts are indicated with "i", followed by the corresponding chromosome number."

The authors might like to comment on the contrast between their findings regarding ATRX depletion in human ES cells and what is seen in mouse ES cells:

We thank the reviewer for highlighting these differences between our findings in hESCs and previous findings in mouse embryonic stem cells (mESCs) as in the work of Wong HL et al; 2010 (doi:10.1101/gr.101477.109, reference number 12). The key differences are the use of mouse cells and the technique of targeting ATRX. While we appreciate the findings supported in this previous work in mESCs, the authors use siRNAs to deplete ATRX transcripts. Although being a powerful tool for the observation of acute changes in the cells, siRNAs result in only transient perturbations of the gene product and cannot be fully compared to the deletion of the endogenous locus, as in our work. The phenotypes that we present in our manuscript are the consequences of a total and constant absence of the ATRX gene product. Despite the technical differences, our findings partially recapitulate some of the phenotypes observed in mESCs. Wong HL et al. report a slight decrease in cell growth at 48 hours upon ATRX knockdown as we also observe in our hESCs after loss of conditional TERT (see Supplementary Fig. 3D and 3I). Our data also show an increase in TIFs in ATRX^{-/-} cells (see Fig. 1G and Supplementary Fig. 1I), although this is only present upon differentiation as we do not detect an appreciable increase in TIFs in ATRX^{-/-} hESCs when compared to the control ATRX^{+/+} hESCs. We cannot exclude that differences in the embryonic stage between mESCs and our hESC ALT model might account for the dissimilarities between the two models. Also, major differences in the telomere maintenance between murine and human have been thoroughly investigated over the years. We have added a sentence to the discussion to highlight these differences and similarities.

Discussion add a sentence that can explain this. Mouse vs human, RNAi vs Ko. Then not fully inconsistent, Acute vs long-term.

Reviewer #2 (Remarks to the Author):

In this manuscript the Hockemeyer group focuses on the requirements for ALT activation, using a novel and unique system, where gene targeting in human pluripotent stem cells is employed to test ALT requirements upon stem cell differentiation. ATRX deficient p16 negative hESC were differentiated into fibroblasts, whereupon transduction with SV40LT allowed about 1% of the cells to survive and form clones that exhibited all recognized signs of ALT. Alt induction was dependent on ATRX, but continued proliferation required p53 suppression in addition. Differentiation was a required step in the ALT activation program, since p16/p53/ATRX negative hESC did not acquire ALT phenotypes when telomere crisis was initiated as consequence of telomerase deletion. Remarkably, telomerase expression was fully compatible with ALT activation upon differentiation. Differentiation led to the establishment of immortality via ALT, since removal of telomerase did not impact long-term proliferation. Sophisticated genome analysis revealed that ATRX loss does not lead to major genome instability, but ongoing ALT subsequently does. Instability was instigated by telomere fusions, which were highlighted by Micro-C. In conclusion, the authors suggest a novel model, where ATRX loss generates an epigenetic environment conducive of ALT activation.

In summary, this is a thorough and novel study evaluating ALT activation, using an innovative and powerful model. Several discoveries render the approach here of high interest to a wide readership:

- AKT activation is independent of telomere crisis
- P53 loss is required for ALT dependent long-term growth
- Telomerase expression is not incompatible with ALT activation
- Accumulation of genome instability is initiated by end-to-end fusion

As such, the manuscript is undoubtedly suitable for Nature Communications and will be a highly cited novel contribution to the field.

A few points should be addressed in a revision. I understand that some of these are complicated and might exceed the scope of the manuscript:

1) What does the loss of CDKN2A/p16 do in the early stages of ALT activation? If I understand this correctly (authors, please indicate if I am wrong), ATRX deletion in hESC does not lead to viable cells unless p16 is co-deleted. Is it possible that p16 loss contributes to the epigenetic environment that allows ATRX to establish ALT predisposition? Similarly, are the hESC in Fig2 also p16 deficient?

We apologize for not being clearer about the different genotypes used in these experiments. The experiments shown in Figures 1 and 2 are deficient for exon 2 of CDKN2A. Targeting of exon 2, shared between p14 and p16 genes, leads to the knockout of both genes. Thus, the resulting cell lines will be both p14^{-/-} and p16^{-/-}, these are indicated in the text as CDKN2A^{-/-}. In order to make our gene targeting strategy clearer we have now modified Supplementary Fig. 1 and we report in more detail the targeting of the CDKN2A locus. We are also more specific in the figure legends and the text for Supplementary Figure 1A now reads “(A) Schematic of endogenous ATRX knockout (ATRX^{-/-}) in a CDKN2A knockout genetic background (CDKN2A^{-/-}) (see Methods). Blue bars represent exons to scale, sgRNAs are indicated in red. Deletion of shared exon 2 (E2) in the CDKN2A locus leads to loss of both p14 and p16. Excision of exon 1 (E1) sequence between sgRNAs in the ATRX locus removes transcription and translation start sites.”

This data argues that in our cell system SV40 LT overexpression is more penetrant in inactivating p53 than the indirect effect of p14 loss. This is consistent with the experiment in Figure 2G, which shows that the direct ko of p53 leads to the same outcome as expression of SV40 LT.

The isogenic ATRX^{+/+} TERT^{c/c} and ATRX^{-/-} TERT^{c/c} cells used in Figures 3 to 6 carry an inactivating insertion in p53 and a homozygous deletion for exon E1a in CDKN2A. Deletion of this exclusive exon leads to genetic loss of p16, but not p14.

2) What happens to the 99% of cells that don't survive in the Fig 1 experiment? I understand that this might be hard to impossible to evaluate experimentally, but it would be interesting the read the authors' view here. Is it possible that the cell death resembles a crisis event?

The reviewer raises an interesting point that we try to address in Figure 2 where we analyze the cells shortly after differentiation. Already at this timepoint cells present DNA damage, EdU incorporation at telomeres and telomere length heterogeneity. Consistent with the experiments shown in Figure 1, the cells arrest and only loss of p53 allows the bypass of this arrest. This suggests to us that differentiation without ATRX being present triggers a DNA damage response at telomeres and aberrant repair. We speculate that either the SV40 LT expression or the loss of p53 is sufficient to bypass the cellular arrest but that the resulting DNA aberrations are detrimental to the cells. To summarize, we fully agree with the reviewer that the cell death resembles a crisis event, just one that is not triggered by cells continually shortening their telomeres given the very short time frame of the arrest. We have now included a sentence in the discussion to address this hypothesis.

3) Is it possible to add stats/error bars to the c-circle experiments?

Yes, we appreciate the reviewers comment and would like to direct attention to the data in figure 3G (former Supplementary Figure 3F). It turns out that the quantity of C-circles detected within different clones is variable even within the same biological replicates. We include data in Supplementary Figure 3H that reflects this and now point this out in the text more carefully.

4) Many established ALT lines have lost cGAS/STING signaling, presumable to prevent the extrachromosomal DNA to activate innate immune signaling. Is this pathway still active in the immortal ALT cells here? Is there a high degree of phosphor TBK signaling?

Thank you for this excellent suggestion. Our previous attempts to directly read-out cGAS/STING had failed, but as suggested by the reviewer we analyzed phospho-TBK1. Indeed, phospho -TBK1 seems specifically attenuated in ATRX deficient cells. We now include this data as Supplementary Figure 7F.

5) Is it possible to tell whether telomerase is active in the ALT cells where TERT has not been deleted? In other words, is it possible to tell whether recombination exists in parallel with telomerase-based telomeric nucleotide incorporation? Similarly, do the authors know whether residual TERT activity in ATRX negative ALT positive cells prevents genome instability accumulation, presumably by preventing some of the initial end to end fusions?

We thank the reviewer for the comment. In our work we show that both TERT^{c/c} and TERT^{-/-} cells show the ALT phenotypes once differentiated and in the ATRX^{-/-} genetic background (see Figure 3B-F). In addition to the presence of APBs and EdU incorporation, ATRX^{-/-} TERT^{c/c} fibroblasts also present heterogeneous telomeres (Figure 3J). As we suggest in the text, this would support the conclusion that even in the presence of telomerase overexpression ATRX^{-/-} cells would trigger ALT upon differentiation. As specifically suggested by the reviewer we have verified that there is no residual telomerase activity after Cre loopout of TERT from the AAVS1 locus. We have performed an *in vitro* telomerase repeat amplification assay (TRAP) that we now include as Supplementary Figure 4K. As expected, the assay shows telomerase activity in TERT^{c/c} fibroblasts and absence of activity upon Cre loopout in both ATRX^{+/+} and ATRX^{-/-} genotypes. To strengthen our hypothesis that the presence of telomerase is compatible with the appearance of ALT phenotypes we now also included a TRF blot as Supplementary Figure 4J. This blot shows that the ATRX^{-/-} cells present heterogeneous lengths of telomeres in both TERT^{c/c} and TERT^{-/-} background after differentiation. This suggests that despite the presence of telomerase elongation, the loss of ATRX and subsequent differentiation still contribute to the appearance of heterogeneous telomeres over time.

6) Please remove the word 'significant' from page 8, second paragraph, last sentence, unless backed up by stats.

We appreciate the clarification from the reviewer, and we have now modified the text in order to more precisely reflect the data presented. The text now reads : "We conclude that ATRX^{-/-} in hESCs does not cause major intra-genomic instability, but that activation of ALT following differentiation results in a detectable amount of genome rearrangement."

Reviewer #3 (Remarks to the Author):

Turkalo et al present evidence that induction of ALT is modulated not by telomere status (length or damage), but rather by cellular state. The authors utilize the differentiation of human embryonic stem cells to reach this conclusion, showing that despite ATRX loss (in different genetic

backgrounds and different telomere lengths), hESCs do not show several molecular characteristics usually associated with ALT. This immediately changes when the authors differentiate these cells to fibroblasts (and neural progenitor cells), where, regardless of telomere damage, loss of ATRX promotes these molecular markers of ALT.

While I agree this phenotype is interesting, the authors keep claiming (including in their title), that an epigenetic switch controls this change. The authors actually claim that they “demonstrate that an epigenetic switch occurs in ATRX-negative stem cells upon differentiation that allows for telomere recombination and cellular immortalization”. No evidence at all is presented for this. No experimental support is presented for such a bold suggestion. No data presenting any sort of epigenetic analysis is shown. This significantly hampers enthusiasm with this manuscript.

We thank the reviewer for their positive and very constructive feedback.

In regard to using the term epigenetics in the discussion and title, we believe that in its most stringent definition we do use the term epigenetics correctly. The central finding of our paper is that a change in the differentiation status of genetically identical cells is sufficient to trigger ALT phenotypes. We believe our evidence supports the finding that the trigger is an epigenetic switch, considering the rapid appearance of these phenotypes and the direct correlation with the differentiation status. We agree with the reviewer that we do not know the molecular underpinning of this epigenetic switch, however, a mechanistic understanding of how the loss of ARTX function and presumably the deposition of H3.3 causes telomere recombination is beyond the scope of the presented work.

Technically, the manuscript could be improved, as p-values are missing, figure legends are poorly described and the amount of data presented (sometimes repetitive) makes this an extremely challenging manuscript to read. The quality of images for some of the IFs (see below) and the light microscopy (1A and 4A) should be improved.

We thank the reviewer for pointing this out. We have replaced the pictures in Figure 4 with better images and improved the contrast for the images shown in Figure 1.

Additionally, the authors make several bold claims on how their data correlates with what is observed with ALT in tumors, again without presenting any evidence for that. This makes the text unnecessarily long and at times reading more like a review than a primary manuscript (this is particularly the case in the Discussion).

After the suggestion by the reviewer, we now provide a shorter version of the discussion.

Some more specific comments:

Figure 1:

- Legend says “Only ATRX^{-/-} proliferated after SV40 transduction”. But this is not the case, as ATRX^{+/+} fibroblasts normally proliferate (Sup. Figure 1E). Do the authors mean: ATRX^{-/-} fibroblasts only proliferate after SV40 transduction?

We thank the reviewer for pointing this out. We have now corrected figure legends to better represent the data shown in the panel. The Figure legend now reads “(A) Experimental overview of ATRX^{-/-} hESCs differentiation and SV40-LT immortalization. ATRX was genetically ablated in

hESCs, then cells were differentiated into fibroblasts (see Methods). Early after differentiation (20 days) cells were either mock transduced or infected with RFP-SV40 LT. Transduced cells proliferated and the efficiency of survival was estimated by counting surviving fibroblast colonies.”

- Authors should show that ATRX^{+/+} fibroblasts (with and without SV40) do not show APBs

We appreciate the reviewer’s comment and now show this data as an additional control included in Supplementary Figure 1G-H.

- TRF experiments are not convincing to show ALT (heterogeneity could have come from differentiation). A negative control (ATRX^{+/+}) should be added (different fibroblast clones).

We thank the reviewer for pointing this out. However, since ATRX^{+/+} fibroblasts were not obtained simultaneously we did not consider including these in the same blot. To strengthen our argument , we provide a TRF blot in Fig. 2F comparing telomere lengths in both ATRX^{+/+} and ATRX^{-/-} cells in both hESC or differentiated states. The blot is consistent with all other TRF analysis showing that only ATRX^{-/-} cells present heterogeneous telomere lengths after differentiation.

- Figures 1D-G: why are authors comparing with HeLa, if they have an isogenic, fibroblast control (ATRX^{+/+})? These cells would be a much better control. (HeLa cells could be kept of course, as these are non-ALT).

Please see our response above. We now include data that show that ATRX^{+/+} fibroblasts are negative for APBs and ATRX^{-/-} are positive; please see Supplementary Figure 1G and 1H.

As for the data shown in Figure 1, ATRX^{+/+} fibroblasts were not obtained simultaneously to their ATRX^{-/-} counterparts. The isogenic hESC ATRX^{+/+} cell line is instead included in the TRF panel in Figure 1 in order to highlight the telomere length setpoint.

- Figure 1F: what is the additional “neg” sample shown specifically on that figure (not described in the legend)

We thank the reviewer for noting this erroneous label and we have now modified the text with the correct label for the control sample (water). We have also corrected the corresponding figure legend 1F that now reads “(F) Quantification of C-circle assay. The assay was performed in HeLa and U2OS cells as negative and positive controls respectively (black bar). In addition, H₂O is used as reaction negative control (indicated as “neg”). ATRX^{-/-} bulk fibroblast culture and a representative clonal line (red bars) show an increase in C-circle signal when compared to controls. C-circles signal intensity was detected after hybridization (see Methods), corrected on Alu signal intensity and normalized for U2OS signal.”

Figure 2:

- Images shown in 2A are not convincing. ATRX^{-/-} fibroblasts only have 2 TRF1 foci (this should be specifically quantified), and several foci in ATRX^{-/-} hESC seem to colocalize. More representative images should be shown.

- Figure 2F: telomeres seem heterogeneous and elongated also in in ATRX^{-/-} hESC, not only ATR^{-/-} fibroblasts.

The ATRX^{+/+} and ATRX^{-/-} cells each have a specific telomere length set-point and telomere length distribution, and telomere length changes are best interpreted within each clone. The key comparison in these experiments is that in ATRX^{-/-} deficient cells telomeres become more heterogeneous shortly after differentiation, an effect that is not seen in the ATRX^{+/+} cells.

- Figure 2D: c-circle images should be shown, not only quantification. And again, it does seem like ATRX^{-/-} hESC have an increase? Also, how many times have c-DNA analysis been performed?

We thank the reviewer for helping improve this data. We now provide an image of the c-circle assay presented in Figure 2D (see Supplementary Figure 1J). Moreover, as requested by the reviewer in the comment below, we now include c-circle data for several ATRX^{-/-} and isogenic ATRX^{+/+} controls in the main figure. Please see below.

- Is p53 activated in ATRX^{-/-} fibroblasts and hESCs? This should be shown. Also, authors should show comet assays for DNA damage analysis in both cell types and both genotypes.

While we have not directly analyzed p53 protein abundance, our genetic experiments in Figure 2G functionally test this and corroborates the effect of SV40LT overexpression. As for the comet assay, we feel that we provide several lines of evidence for the genomic instability of differentiated ATRX cells including strand-seq, metaphase spreads and telomere FISH as well as Hi-C data. These assays assess instability both genome-wide and more specifically at the telomeres. If any additional insights could be gained from the Comet assay these experiments are beyond the scope of this study.

Figure 3:

- Figures 3 and Sup. 3 are interesting but extremely confusing. Some of it is understandable, as the authors show many different genotypes in different cellular states. But some of it could be made better, which would facilitate data interpretation. Figure 3B is sub-par. C-circles should be incorporated to main figure, and again, they seem to show that hESCs do show c-circles upon ATRX deletion? Authors should compared side-by-side (hESC vs fibroblast, different genotypes)

Yes, the reviewer is correct that C-circles accumulate more in ATRX^{-/-} hESCs that are p53 deficient. As discussed, this accumulation seems to require p53 loss of function and persists in differentiated cells. As to the presence of c-circles in the undifferentiated state, we now reference a key study by the Karlseder lab that explicitly investigates the presence of c-circles in hESCs and discuss this in the content of our findings.

We agree with the reviewer that the key finding of our experiments is that c-circles accumulation in hESCs increases upon ATRX deletion in TERT^{cl} cells. As suggested by the reviewer we have moved this data into the main figure (see Figure 3G).

- SF3H: why would the deletion of TERT immediately cause a slow in proliferation in ATR^{-/-} hESCs and not in ATRX positive hESCs?

The reviewer raises an interesting question. We think that this is either the consequence of the shorter telomere length set point of ATRX^{-/-} hESCs or an unknown mechanism by which telomerase can partially compensate for the loss of ATRX in the stem cell state.

Figure 5:

- Supplementary Table 4 should be added to Figure 5 for clarity.

We appreciate this suggestion, and the Table has now been moved to the main figure as Table 1.

Figure 6:

- U2OS cells should be added as control to 6B-C

We thank the reviewer for this suggestion. We now present a new panel as Figure 6C that includes U2OS and HeLa 1.3 cells as positive and negative controls respectively. The graph represents two independent experiments.

- On SF6H, the difference on SCEs us impossible to visualize. 1) Why? Shouldn't the ATRX^{-/-} cells have a significantly higher number of SCEs? 2) Despite this minimal difference, the authors show significance. The graph should be "broken" in two, at different scales, for differences to be visible.

We thank the reviewer for this helpful comment. Supplementary figures 6F and 6H have now been modified accordingly . The left Y axis is now split in two, helping to better appreciate the differences among the different samples. Statistical analysis according to Kruskal-Wallis test are now reported in the figure panels.

Finally:

- It would be interesting to perform the opposite experiment: if a panel of ALT positive cells is reprogrammed back to a pluripotent state, is ALT silenced upon induction of pluripotency?

This is an excellent suggestion and I (Dirk Hockemeyer during my postdoc) have attempted this experiment for several ALT cell lines as well as other cancer cell lines. It turns out that reprogramming of cancer cells is exceedingly difficult as the cells do not revert to an epigenetically stable stem cell state that can be called pluripotent or where cells have a predefined cell morphology and expression profile. This is consistent with published mouse data (PMID: 15289459).

REVIEWER COMMENTS

Reviewer #1 (Remarks to the Author):

The authors have addressed my comments satisfactorily.

There are still a couple of errors that need correcting:

Figure 3H should be 3J.
Supplementary Fig 3I still shows ATRX KO not TERT KO.

Reviewer #2 (Remarks to the Author):

I thank the authors for their clarifications and the changes made. My concerns have been satisfied and I support publication of the manuscript in the current form.

Reviewer #3 (Remarks to the Author):

The manuscript has improved, and the editing errors (missing information, wrong labels, lack of stats, poor quality figures) mentioned by both Reviewers #1 and #3 have been addressed.

It is however somewhat frustrating that some of the requests were classified by the author as “outside the scope of this manuscript”, or referred back to unsuccessful experiments performed by the PI several years ago, while still a postdoc. Reprogramming efficiencies have greatly improved, new reprogramming methods have become available, and it could be that U2OS or other ALT positive cell lines would successfully be reprogrammed. The authors could have tried to perform those experiments, and it is frustrating that it didn't happen.

While I believe this is an interesting manuscript, there is still one major point that the authors and I don't see eye-to-eye on:

- To have epigenetic switch in the title, implying this is the reason for ALT only becoming activated in differentiated cells. Not tested at all by the PI. Gene expression profiles are completely different in ES vs differentiated cells (and I agree there is a clear epigenetic aspect to that), metabolism rates are different, etc. To make a strong argument like that in the title, and simply not look at it all in the entire manuscript, is misleading. And it is unnecessary. What the authors show is that cell state is a factor in ALT triggering. Which is also interesting.

REVIEWER COMMENTS

Reviewer #1 (Remarks to the Author):

The authors have addressed my comments satisfactorily. There are still a couple of errors that need correcting: Figure 3H should be 3J. Supplementary Fig 3I still shows ATRX KO not TERT KO.

We thank the reviewer for their help improving our manuscript. We greatly appreciate their time and effort. We have now made these changes.

Reviewer #2 (Remarks to the Author):

I thank the authors for their clarifications and the changes made. My concerns have been satisfied and I support publication of the manuscript in the current form.

Thank you for your constructive help improving our work!

Reviewer #3 (Remarks to the Author):

The manuscript has improved, and the editing errors (missing information, wrong labels, lack of stats, poor quality figures) mentioned by both Reviewers #1 and #3 have been addressed.

We would like to thank the reviewers for their additional time and comments to make our paper better. Their suggestions greatly improved the presentation of our work.

It is however somewhat frustrating that some of the requests were classified by the author as “outside the scope of this manuscript”, or referred back to unsuccessful experiments performed by the PI several years ago, while still a postdoc. Reprogramming efficiencies have greatly improved, new reprogramming methods have become available, and it could be that U2OS or other ALT positive cell lines would successfully be reprogrammed. The authors could have tried to perform those experiments, and it is frustrating that it didn't happen.

We apologize that we did not explain our motivation better. We did not perform additional experiments to reprogram ALT cancer cells as we are unsure how to interpret the potential outcomes of these experiments. There could be a number of reasons – many of which could be independent of the ALT process – why these cells would not be reprogrammable or why ALT cannot be repressed. Moreover, if we were to find that some cell clones after “reprogramming” would lose some features of ALT, the genetic and epigenetic heterogeneity of ALT cells would make it effectively impossible to determine if this was due to an epigenetic change induced by reprogramming factors or due to preexisting or induced genetic changes. A key finding of our work is that ALT can be induced upon differentiation of ATRX deficient hESCs. We show that in our system ALT features are rapidly induced within a large population of cells without selecting for additional mutations.

The low efficiency and extended proliferation of putative reprogrammed cancer cells is not a suitable tool to address if the inverse is true.

While I believe this is an interesting manuscript, there is still one major point that the authors and I don't see eye-to-eye on:

- To have epigenetic switch in the title, implying this is the reason for ALT only becoming activated in differentiated cells. Not tested at all by the PI. Gene expression profiles are completely different in ES vs differentiated cells (and I agree there is a clear epigenetic aspect to that), metabolism rates are different, etc. To make a strong argument like that in the title, and simply not look at it all in the entire manuscript, is misleading. And it is unnecessary. What the authors show is that cell state is a factor in ALT triggering. Which is also interesting.

From the reviewer's additional comments, we now have realized that our definition of 'epigenetic switch' could be misleading to the reader. We use the term under the broadest definition: "an inheritable trait that is not caused by changes in the nucleotide sequence of the genome." It is clear that this is too broad and that our title and discussion imply that we have identified the molecular switch that underlies the induction of ALT features in our cell system. We have not and future experiments are needed to do so. We have changed the title and the text to better reflect what we are showing and what not and try to avoid whenever possible the term epigenetics.